# ONLINE CONTINUAL LEARNING UNDER REAL CONCEPT DRIFT: A STATISTICAL PERSPECTIVE

## ABSTRACT

Real-world data often exhibit non-stationarity, prompting growing interest in adaptive learning techniques. Continual learning, which aims to sequentially learn multiple tasks, provides a promising framework to address this challenge. However, learning under real concept drift, where the relationship between inputs and outputs evolves over time, remains relatively underexplored. In this paper, we propose a novel regularization-based method that incorporates a memory buffer to improve robustness against concept drift. Assuming the existence of a common center for the evolving true models, our method jointly constrains current and past task estimates, effectively bridging them to form a stable estimate that incorporates information across tasks. To further adapt to task variability, we develop an online algorithm that dynamically tunes task-specific regularization parameters. We also provide theoretical guarantees by deriving an error bound that characterizes the overall performance of the estimator, explicitly capturing the effects of task-relatedness, memory buffer size, and regularization strength. Extensive experiments demonstrate that our method achieves superior stability–plasticity trade-offs under varying degrees of task similarity.

## 1 INTRODUCTION

Classic online learning algorithms typically assume that data are generated from a stationary probability distribution and arrive sequentially over time (Robbins and Monro, 1951; Duchi et al., 2011; Shalev-Shwartz et al., 2012; Kingma and Ba, 2014; Luo and Song, 2020). However, this assumption is often violated in real-world applications, where data streams are inherently non-stationary. Directly applying these algorithms to non-stationary environments leads to catastrophic forgetting (McCloskey and Cohen, 1989), a phenomenon in which the model rapidly forgets previously acquired knowledge when adapting to new data. Although several extensions of online learning have been proposed to handle distributional shifts (Dekel et al., 2006; Zhang et al., 2018), they primarily focused on rapid adaptation to recent observations and typically struggle to retain long-term information. To overcome this limitation, the paradigm of continual learning (CL), also referred to as lifelong learning, has emerged as a promising direction. It mimics the lifelong learning ability of humans by enabling the model to learn continuously from non-stationary data while retaining long-term knowledge.

A more comprehensive view of CL is the stability-plasticity dilemma (Mermillod et al., 2013), wherein stability refers to the preservation of previously acquired knowledge, and plasticity denotes the ability to quickly adapt to new information. Effective CL aims to strike a balance between these two competing objectives. To this end, a variety of algorithms have been proposed in the literature, which can be grouped into three main categories: (1) *Regularization-based methods*, which introduce constraints on parameter updates to prevent the overwriting of previously learned representations, thereby reducing catastrophic forgetting (Kirkpatrick et al., 2017; Zenke et al., 2017; Aljundi et al., 2018; Heckel, 2022); (2) *Expansion-based methods*, which preserve prior knowledge by freezing important weights and expanding the network architecture when encountering new tasks (Rusu et al., 2016; Yoon et al., 2017; Li et al., 2019); (3) *Memory-based methods*, which store samples from past data for replay or regularization, helping to maintain performance on earlier tasks during new task learning (Rebuffi et al., 2017; Lopez-Paz and Ranzato, 2017; Chaudhry et al., 2018; Rolnick et al., 2019). In this work, we focus on structure-preserving approaches, particularly those

based on regularization and memory replay, as they offer a favorable trade-off between scalability and the retention of accumulated knowledge in dynamic learning environments.

One of the key challenges in such environments is the presence of non-stationary data streams. Designing effective CL algorithms requires a precise understanding of how the data stream evolves. Recent efforts have focused on two common forms of distributional shift: domain-incremental, where the input distribution changes over time, and task- or class-incremental, where new labels are introduced sequentially (Schwarz et al., 2018; Aljundi et al., 2019a; Cai et al., 2021; Li et al., 2023; Ghunaim et al., 2023; Verwimp et al., 2023). In contrast, relatively limited attention in CL has been given to real concept drift, a more subtle yet practically important scenario in which the underlying relationship between inputs and labels changes while the distribution of input may stay unchanged (Lesort et al., 2021). Our work is to address CL under real concept drift. To clearly delineate our goal, we now present a formal problem formulation.

**Problem formulation**    We consider a sequence of tasks indexed by $t = 1, \ldots, T$, where each task draws data from an unknown distribution $\mathcal{D}_t$. Specifically, for task $t$, we observe a dataset $z_t \sim \mathcal{D}_t := \{(\mathbf{x}_t^i, y_t^i) \in \mathbb{R}^p \times \mathbb{R}\}_{i=1}^{n_t}$, where $n_t$ denotes the number of observations. Let $\mathcal{W} \in \mathbb{R}^p$ be the parameter space, and define $\ell_t(\boldsymbol{\omega}, z_t)$ as the task-specific loss on data $z_t$. To guarantee the generalization error of a CL algorithm, a straightforward way is to minimize the average population loss by pooling all tasks together:

$$\frac{1}{T} \sum_{t=1}^{T} \mathcal{L}_t(\boldsymbol{\omega}) := \frac{1}{T} \sum_{t=1}^{T} \mathbb{E}_{z_t \sim \mathcal{D}_t} \ell_t(\boldsymbol{\omega}, z_t). \tag{1}$$

The non-stationary type considered in this work is called real concept drift, where $\mathbb{P}_t(y|\mathbf{x}) \neq \mathbb{P}_{t+1}(y|\mathbf{x})$ while $\mathbb{P}_t(\mathbf{x}) = \mathbb{P}_{t+1}(\mathbf{x})$. This implies that the underlying relationship between input features and target outcomes evolves over time. For example, in a parametric regression setting where $\boldsymbol{\omega}_t^*$ denotes the true model parameter for task $t$, concept drift manifests as $\boldsymbol{\omega}_t^* \neq \boldsymbol{\omega}_{t+1}^*$.

Such drift is commonly observed in real-world applications (Gama et al., 2014). In recommendation systems, user preferences and behaviors may evolve over time, altering the mapping between user activity and suggested items. Similarly, in financial and environmental domains, external factors such as market shocks, regulatory changes, or climatic events can shift the predictive relationship without affecting the input distribution. The goal of this work is to develop a CL framework that adapts effectively to these evolving task distributions, while preserving knowledge from previous tasks and mitigating catastrophic forgetting. Our main contributions are summarized as follows.

- We propose a novel and robust regularization-based method equipped with a memory buffer to address concept drift in CL. By jointly constraining the loss functions from both current and previous tasks, our approach effectively integrates information over time to produce stable and adaptive model updates. Moreover, our algorithm is computationally efficient, performing online updates within each task without requiring costly batch retraining.

- Most existing theoretical analyses in CL are limited to linear models or assume covariate shift. In contrast, we establish high probability generalization error bounds for our proposed estimator under a more general setting. Our analysis explicitly quantifies the influence of factors such as task relatedness, memory buffer size, and regularization strength on learning performance.

- We conduct extensive numerical comparisons with state-of-the-art methods using both synthetic datasets, spanning low- and high-dimensional regimes and a real-world benchmark. The results demonstrate that our approach consistently achieves superior robustness, predictive accuracy, and computational efficiency.

**Notations**    We use the symbol $[n]$ as a shorthand for $\{1, 2, \cdots, n\}$ and $|\cdot|$ to denote the absolute value of a real number or cardinality of a set. For nonnegative sequences $\{a_n\}_{n=1}^{\infty}$ and $\{b_n\}_{n=1}^{\infty}$, we write $a_n \lesssim b_n$ if there exists a positive constant $C$ such that $a_n \leq C b_n$. In addition, we write $a_n \asymp b_n$ if $a_n \lesssim b_n$ and $b_n \lesssim a_n$. Let $\{e_j\}_{j=1}^p$ denote the canonical bases of $\mathbb{R}^p$. Define $\mathbb{S}^{p-1} = \{\mathbf{x} \in \mathbb{R}^p : \|\mathbf{x}\|_2 = 1\}$ and $\mathcal{B}(\mathbf{x}, r) = \{\boldsymbol{y} \in \mathbb{R}^p : \|\boldsymbol{y} - \mathbf{x}\|_2 \leq r\}$ for $\mathbf{x} \in \mathbb{R}^p$ and $r \geq 0$. Define $\|X\|_{\psi_2} = \sup_{q \geq 1}\{q^{-1/2}\mathbb{E}^{1/q}|X|^q\}$ and $\|X\|_{\psi_1} = \sup_{q \geq 1}\{q^{-1}\mathbb{E}^{1/q}|X|^q\}$ for a random variable $X$; $\|\mathbf{x}\|_{\psi_2} = \sup_{\|\boldsymbol{\omega}\|_2 = 1} \|\langle \boldsymbol{\omega}, \mathbf{x} \rangle\|_{\psi_2}$ for a random vector $\mathbf{x}$.

## 2 RELATED WORK

In this section, we review related work in three key areas: online CL, online multi-task learning, and the theoretical foundations of CL.

**Online continual learning**   Unlike traditional CL, which follows a batch learning paradigm, online CL assumes that data arrive sequentially or in small batches, and that previously seen data from either the current or past tasks are no longer accessible. This setting poses significant challenges for mitigating catastrophic forgetting. To address this, Lopez-Paz and Ranzato (2017) proposed the widely cited Gradient Episodic Memory (GEM), which formulates a constrained optimization problem using an episodic memory buffer to ensure non-negative backward transfer. Subsequently, Chaudhry et al. (2019) examined the performance of CL methods under severely restricted memory budgets, demonstrating the surprising effectiveness of small memory buffers. Building on this, several methods have focused on selecting representative samples for storage in memory. For example, Aljundi et al. (2019b) introduced a gradient-based sample selection strategy, while Sun et al. (2022) developed an information-theoretic approach to optimize memory usage. Although these methods assume fixed, limited memory, Prabhu et al. (2023) extended the setting to relaxed storage constraints under limited computational resources, further broadening the scope of online CL.

**Online multi-task learning**   CL is closely related to online multi-task learning (OMTL), as both frameworks address sequential data arising from multiple tasks and aim to accumulate knowledge over time. However, their objectives are fundamentally different: CL emphasizes the continual adaptation of a single model while preserving performance on previously encountered tasks, whereas OMTL focuses on leveraging shared structure across tasks to enhance estimation for each individual task. For instance, Duan and Wang (2023) proposed a class of adaptive estimators that automatically exploit latent task similarities while accounting for task-specific heterogeneity, though their approach is confined to the offline setting. In the online context, Cavallanti et al. (2010) developed Perceptron-based algorithms for multi-task binary classification that incorporate inter-task relationships. To relax the assumption of fixed task dependencies, Saha et al. (2011) introduced an adaptive framework that learns the task interaction matrix directly from data. More aligned with the goals of CL, Ruvolo and Eaton (2013) proposed an efficient lifelong learning algorithm that integrates principles from both transfer learning and multi-task learning. A comprehensive survey of recent advances in OMTL can be found in Zhang and Yang (2021).

**Theory in continual learning**   Recent years have seen growing interest in the theoretical underpinnings of CL, particularly in understanding generalization and forgetting. Bennani et al. (2020) investigated the generalization error and forgetting dynamics of orthogonal gradient descent. Raghavan and Balaprakash (2021) formalizes CL as a trade-off between generalization and forgetting, proves the existence and stability of a balance point via a game-theoretic framework. Lin et al. (2023) further contributed to this literature by deriving the first explicit expressions for expected forgetting and generalization error in general CL settings under over-parameterized linear models. A number of works focus more specifically on linear models. Evron et al. (2023) analyzes continual linear classification on separable data, showing that weakly regularized training reduces to sequential max-margin projections and deriving bounds on forgetting. Goldfarb et al. (2024) analyzes the joint effect of task similarity and over-parameterization on forgetting through a two-task continual linear regression model. Ding et al. (2024) developed a unified theoretical framework for characterizing forgetting in linear regression models trained via stochastic gradient descent, applicable to both under- and over-parameterized regimes. Li et al. (2023) studied domain-incremental CL involving two linear regression tasks in a fixed design setting, and theoretically characterized the trade-off between forgetting and intransigence under an $\ell_2$-regularization scheme. Zhao et al. (2024) extended this analysis to a sequence of linear regression tasks under covariate shift.

## 3 METHODOLOGY

### 3.1 ONLINE ADAPTIVE CONTINUAL LEARNING

We now describe our online adaptive CL framework. Effective knowledge transfer across tasks is contingent upon the presence of sufficient similarity among them. In settings where such relatedness

is absent, learning tasks jointly may result in negative transfer, and task-wise independent learning becomes preferable. Let $\boldsymbol{\omega}_{1:T}^* = (\boldsymbol{\omega}_1^*, \ldots, \boldsymbol{\omega}_T^*) \in \mathbb{R}^{p \times T}$ denote the collection of true parameter vectors for all $T$ tasks. To formalize task-relatedness, we adopt the following definition inspired by Duan and Wang (2023):

**Assumption 1** (($\varepsilon, \delta$)-related)**.** *For any $\varepsilon \in [0, 1]$ and $\delta \geq 0$, we assume that $\boldsymbol{\omega}_{1:T}^* \in \Omega(\varepsilon, \delta)$, where*

$$\Omega(\varepsilon, \delta) = \left\{ \boldsymbol{\omega}_{1:T} \in \mathbb{R}^{p \times T} : \min_{\boldsymbol{\omega}_0 \in \mathbb{R}^p} \max_{j \in J} |\boldsymbol{\omega}_j - \boldsymbol{\omega}_0| \leq \delta \text{ and } |J^c|/T \leq \varepsilon \text{ for some } J \subseteq [T] \right\}.$$

Under Assumption 1, the $T$ tasks are said to be ($\varepsilon, \delta$)-related. Here, $\boldsymbol{\omega}_0$ serves as a latent central model around which the majority of task parameters are clustered. The parameter $\delta$ controls the maximum deviation of the related task parameters from this center, while $\varepsilon$ specifies the proportion of tasks that may deviate arbitrarily. Notably, any sequence of $T$ tasks trivially satisfies $(0, \max_{t \in [T]} \|\boldsymbol{\omega}_t^*\|_2)$-relatedness, regardless of structure. Smaller values of $\varepsilon$ and $\delta$ indicate stronger task similarity, with the limiting case $\varepsilon = \delta = 0$ corresponding to the classical stationary setting in which all tasks share the same underlying parameter: $\boldsymbol{\omega}_1^* = \cdots = \boldsymbol{\omega}_T^*$.

To simplify the presentation, we assume that all tasks have the same number of samples, i.e., $n_1 = \cdots = n_T = n$. The empirical loss for the $t$th task is defined as $L_t(\boldsymbol{\omega}) = \sum_{i=1}^n \ell_t(\boldsymbol{\omega}, z_t^i)/n$, where $z_t^i$ denotes the $i$th sample from $t$th task. Suppose the algorithm is equipped with a memory buffer $\mathcal{M}$, subject to a storage budget $M$, which retains a subset of previously observed data to mitigate forgetting. Let $m \leq M$ be the current size of the memory buffer. Define the empirical loss over the memory buffer as:

$$L_{\text{past}}(\boldsymbol{\omega}) = \frac{1}{m} \sum_{(k,i) \in \mathcal{M}} \ell_k(\boldsymbol{\omega}, z_k^i), \quad k = 1, \ldots, t-1,$$

where $z_k^i$ is the $i$th sample from task $k \in \{1, \ldots, t-1\}$, and $(k, i) \in \mathcal{M}$ indexes the stored instances.

To accommodate potential concept drift in CL, we propose the following regularized optimization problem for the $t$th task:

$$\min_{\boldsymbol{\omega}_{\text{past}}, \boldsymbol{\omega}_t, \boldsymbol{\theta} \in \mathbb{R}^p} a_1\{L_{\text{past}}(\boldsymbol{\omega}_{\text{past}}) + \lambda_{\text{past}}\|\boldsymbol{\omega}_{\text{past}} - \boldsymbol{\theta}\|_2\} + a_2\{L_t(\boldsymbol{\omega}_t) + \lambda_t\|\boldsymbol{\omega}_t - \boldsymbol{\theta}\|_2\}, \tag{2}$$

where $\lambda_t, \lambda_{\text{past}}$ are regularization parameters controlling the proximity between task-specific parameters $\boldsymbol{\omega}_t$, $\boldsymbol{\omega}_{\text{past}}$ and the shared latent vector $\boldsymbol{\theta}$, and $a_1, a_2 \geq 0$ are weighting coefficients satisfying $a_1 + a_2 = 1$. Solving equation 2 yields a triplet $(\hat{\boldsymbol{\omega}}_{\text{past}}, \hat{\boldsymbol{\omega}}_t, \hat{\boldsymbol{\theta}})$, where $\hat{\boldsymbol{\omega}}_{\text{past}}$ and $\hat{\boldsymbol{\omega}}_t$ are task-specific solutions for the memory buffer and current task, respectively, while $\hat{\boldsymbol{\theta}}$ serves as a unified estimate capturing shared information across tasks. We adopt $\hat{\boldsymbol{\theta}}$ as the final output of the algorithm after processing task $t$, as it balances both retention of prior knowledge and adaptation to new information.

The formulation in equation 2 unifies our online adaptive memory-based and regularization-based CL paradigms. In the limiting case where $\lambda_{\text{past}}, \lambda_t \to \infty$, the problem reduces to a memory-based approach aligned with experience replay strategies (Riemer et al., 2018; Hayes et al., 2019; Chaudhry et al., 2019). The inclusion of $\ell_2$-regularization terms facilitates controlled sharing of statistical strength across tasks, promoting robustness to concept drift. Specifically, larger regularization parameters shrink the discrepancy between $\hat{\boldsymbol{\omega}}_{\text{past}}$, $\hat{\boldsymbol{\omega}}_t$, and $\hat{\boldsymbol{\theta}}$, enhancing model stability. In contrast, smaller values, particularly a small $\lambda_{\text{past}}$, allow the model to more flexibly accommodate task-specific deviations, favoring plasticity in settings where tasks are weakly related.

**Remark 1.** *The weight parameters $a_1$ and $a_2$ control the trade-off between stability and plasticity in the learning process. A larger value of $a_1$ emphasizes the influence of past knowledge, encouraging the model to preserve previously learned information and promoting stability. In contrast, a larger $a_2$ increases the model's responsiveness to new data, enhancing plasticity. To reflect the relative sizes of the memory buffer and the current task, we adopt the principled choice $a_1 = m/(n+m)$ and $a_2 = n/(n+m)$ throughout our theoretical analysis and experiments.*

**Remark 2.** *In scenarios where the primary focus is on the performance of the current task, such as in online multi-task learning, it is natural to output the task-specific estimate $\hat{\boldsymbol{\omega}}_t$, instead of the shared parameter $\hat{\boldsymbol{\theta}}$. This choice preserves the task-adaptive nature of the estimator while still benefiting from shared information across tasks through the regularization framework.*

To solve the optimization problem in equation 2 in practice, we propose an online adaptive CL procedure in Algorithm 1, which includes key components such as parameter selection, online model training, and memory buffer updates. For computational convenience, we define task-specific corrections as $\boldsymbol{\nu}_t = \boldsymbol{\omega}_t - \boldsymbol{\theta}$ and $\boldsymbol{\nu}_{\text{past}} = \boldsymbol{\omega}_{\text{past}} - \boldsymbol{\theta}$. Details regarding parameter tuning are deferred to Section 3.2. For online memory management, we employ reservoir sampling (see Algorithm 3 in the appendix) to dynamically update the buffer and use uniform random sampling when retrieving stored samples. Although our framework is compatible with more complicated sampling strategies (see Section A.3), we focus on the standard setting in this work to highlight our main methodology.

---

**Algorithm 1** Online Adaptive CL Algorithm

1: **Input:** Data stream $z_t = \{(\mathbf{x}_t^i, y_t^i)\}_{i=1}^n$, memory buffer $\mathcal{M}$, count. Parameters: buffer size $M$, test batch size $B$, candidate set of regularization parameters $\mathcal{S}_\lambda$
2: **Initialize** $\boldsymbol{\nu}_t^0 = \mathbf{0}_p, \boldsymbol{\nu}_{\text{past}}^0 = \mathbf{0}_p, \boldsymbol{\theta}^0 = \hat{\boldsymbol{\theta}}_{t-1}, \mathcal{M}^0 = \mathcal{M}$
3: **for** $i = 1, 2, \ldots, n$ **do**
4:     **if** $i \leq B$ **then**
5:         $(\boldsymbol{\nu}_t^B, \boldsymbol{\nu}_{\text{past}}^B, \boldsymbol{\theta}^B, \lambda_{\text{past}}, \lambda_t) \leftarrow$ DynamicParameterSelection$(z_t^i, B, \mathcal{M}, \mathcal{S}_\lambda)$
6:     **else**
7:         Randomly sample $z_{\text{past}}^i$ from memory buffer $\mathcal{M}$
8:         $\boldsymbol{\nu}_{\text{past}}^i \leftarrow \text{prox}_{\eta_i \lambda_{\text{past}}} \left( \boldsymbol{\nu}_{\text{past}}^{i-1} - \eta_i \nabla \ell_{\text{past},i}(\boldsymbol{\theta}^{i-1} + \boldsymbol{\nu}_{\text{past}}^{i-1}, z_{\text{past}}^i) \right)$
9:         $\boldsymbol{\nu}_t^i \leftarrow \text{prox}_{\eta_i \lambda_t} \left( \boldsymbol{\nu}_t^{i-1} - \eta_i \nabla \ell_{t,i}(\boldsymbol{\theta}^{i-1} + \boldsymbol{\nu}_t^{i-1}, z_t^i) \right)$
10:        $\boldsymbol{\theta}^i \leftarrow \boldsymbol{\theta}^{i-1} - \gamma_{ti} \{a_1 \nabla \ell_{\text{past},i}(\boldsymbol{\theta}^{i-1} + \boldsymbol{\nu}_{\text{past}}^i, z_{\text{past}}^i) + a_2 \nabla \ell_{t,i}(\boldsymbol{\theta}^{i-1} + \boldsymbol{\nu}_t^i, z_t^i)\}$
11:     **end if**
12:     Update memory buffer $(\mathcal{M}^i, \text{count}) \leftarrow$ ReservoirSampling$(z_t^i, \text{count}, \mathcal{M}^{i-1}, M)$
13: **end for**
14: **Output:** $\hat{\boldsymbol{\theta}}_t = \boldsymbol{\theta}^n$, count, and $\mathcal{M} = \mathcal{M}^n$

---

As detailed in Algorithm 1, the model training step (lines 7 to 10) employs a combination of stochastic proximal gradient descent (SPGD) and stochastic gradient descent (SGD) to process data in an online manner. Upon receiving a new data point $z_t^i$, we first fix the shared vector $\boldsymbol{\theta}$ at its previous estimate $\boldsymbol{\theta}^{i-1}$. We then update the task-specific corrections $\boldsymbol{\nu}_{\text{past}}$ and $\boldsymbol{\nu}_t$ using SPGD as follows:

$$
\begin{aligned}
\boldsymbol{\nu}_{\text{past}}^i &= \text{prox}_{\eta_i \lambda_{\text{past}}} \left( \boldsymbol{\nu}_{\text{past}}^{i-1} - \eta_i \nabla \ell_{\text{past},i}(\boldsymbol{\theta}^{i-1} + \boldsymbol{\nu}_{\text{past}}^{i-1}, z_{\text{past}}^i) \right), \\
\boldsymbol{\nu}_t^i &= \text{prox}_{\eta_i \lambda_t} \left( \boldsymbol{\nu}_t^{i-1} - \eta_i \nabla \ell_{t,i}(\boldsymbol{\theta}^{i-1} + \boldsymbol{\nu}_t^{i-1}, z_t^i) \right),
\end{aligned}
\tag{3}
$$

where $\text{prox}_c(\boldsymbol{\omega}) = (1 - c/\|\boldsymbol{\omega}\|_2)_+ \boldsymbol{\omega}$ is the proximal operator, used to handle the $\ell_2$ regularization. The gradient $\nabla \ell_{\text{past},i}$ corresponds to the loss function evaluated at a buffered data point $z_{\text{past}}^i$ sampled randomly from the memory buffer $\mathcal{M}$, while $\nabla \ell_{t,i}$ is evaluated at the current data point $z_t^i = (\mathbf{x}_t^i, y_t^i)$. Once the task-specific corrections are updated, we update the shared parameter $\boldsymbol{\theta}$ using an SGD step while holding $\boldsymbol{\nu}_t^i$ and $\boldsymbol{\nu}_{\text{past}}^i$ fixed:

$$
\boldsymbol{\theta}^i = \boldsymbol{\theta}^{i-1} - \gamma_{ti} \{a_1 \nabla \ell_{\text{past},i}(\boldsymbol{\theta}^{i-1} + \boldsymbol{\nu}_{\text{past}}^i, z_{\text{past}}^i) + a_2 \nabla \ell_{t,i}(\boldsymbol{\theta}^{i-1} + \boldsymbol{\nu}_t^i, z_t^i)\},
\tag{4}
$$

In practice, $\eta_i$ in equation 3 and $\gamma_{ti}$ in equation 4 are both the step sizes, being selected as decreasing sequences to ensure convergence.

## 3.2 DYNAMIC PARAMETER TUNING

In this section, we describe our approach to dynamically selecting regularization parameters in an online fashion. Unlike traditional offline settings, where hyperparameters are commonly tuned via $K$-fold cross-validation using pre-specified training and testing splits, such data partitioning strategies are infeasible in online CL scenarios.

To address this, we propose an adaptive tuning approach for the regularization parameters in the augmented optimization problem equation 2, detailed in Algorithm 2. Let $\mathcal{S}_\lambda$ be a pre-specified set of candidate parameter pairs $(\lambda_t, \lambda_{\text{past}})$, with cardinality $s_\lambda$. For each task $t$, we treat the first $B$ data points as a pseudo-validation set to assess the performance of each candidate pair on both the current task and stored memory buffer data. At the beginning of task $t$ (i.e., at iteration $i = 1$), we initialize a collection of temporary model estimates $\{\tilde{\boldsymbol{\theta}}_k^i\}_{k=1}^{s_\lambda}$, one for each candidate in $\mathcal{S}_\lambda$. As new

data points arrive sequentially, each candidate model is updated online and evaluated on the current data point $z_t^i$ and the memory buffer $\mathcal{M}$ based on prediction error. This results in an $s_\lambda \times (B-1)$ score matrix $\boldsymbol{S}$ by the end of the evaluation window. For example, in linear regression, the optimal parameter index is selected by minimizing the average error over the pseudo-validation set:

$$k_{\mathrm{op}} = \arg\min_{k \in [s_\lambda]} \sum_{i=2}^{B} \frac{1}{2} \{\|y_t^i - \mathbf{x}_t^i \tilde{\boldsymbol{\theta}}_k^{i-1}\|_2^2 + \|y_{\mathrm{past}} - \mathbf{x}_{\mathrm{past}} \tilde{\boldsymbol{\theta}}_k^{i-1}\|_2^2/m\}, \tag{5}$$

where $(\mathbf{x}_{\mathrm{past}}, y_{\mathrm{past}})$ denotes data sampled from the memory buffer $\mathcal{M}$. The final regularization parameters for task $t$ are chosen as $(\lambda_t, \lambda_{\mathrm{past}}) = \{\mathcal{S}_\lambda\}_{k_{\mathrm{op}}}$.

**Remark 3.** *In the evaluation step, e.g., equation equation 5, we assign equal weights to the performance of the candidate estimators $\{\tilde{\boldsymbol{\theta}}_k^i\}_{k=1}^{s_\lambda}$ on both the current task and the memory buffer. This setting reflects a neutral balance between stability and plasticity. However, our framework allows users to tailor the weighting scheme based on specific objectives, as discussed in Remark 1.*

---

**Algorithm 2** Dynamic Parameter Selection

1: **Input:** Data points $\{(\mathbf{x}_t^i, y_t^i)\}_{i=1}^B$, memory buffer $\mathcal{M}$. Parameters: test batch size $B$, candidate set of regularization parameters $\mathcal{S}_\lambda$
2: **Initialize** $\boldsymbol{\nu}_t^0 = \mathbf{0}_p, \boldsymbol{\nu}_{\mathrm{past}}^0 = \mathbf{0}_p, \boldsymbol{\theta}^0 = \hat{\boldsymbol{\theta}}_{t-1}, \boldsymbol{S} = \mathbf{0}_{s_\lambda \times (B-1)}$
3: **for** $i = 1, 2, \ldots, B$ **do**
4:     **if** $i > 1$ **then**
5:         Evaluate $\{\tilde{\boldsymbol{\theta}}_k^{i-1}\}_{k=1}^{s_\lambda}$ on memory buffer $\mathcal{M}$ and current data $z_t^i$, then obtain a score matrix $\boldsymbol{S}_{k,(i-1)}$ for $k \in [s_\lambda]$
6:     **end if**
7:     **for** $k = 1, 2, \ldots, s_\lambda$ **do**
8:         Run lines 7 to 10 in Algorithm 1 and obtain $\tilde{\boldsymbol{\theta}}_k^i := \boldsymbol{\theta}^i$
9:     **end for**
10: **end for**
11: Select parameter pair $(\lambda_t, \lambda_{\mathrm{past}})$ according to the score matrix $\boldsymbol{S}$
12: **Output:** $\lambda_t, \lambda_{\mathrm{past}}, \boldsymbol{\theta}^B, \boldsymbol{\nu}_t^B, \boldsymbol{\nu}_{\mathrm{past}}^B$

---

## 4 THEORETICAL GUARANTEE

In this section, we provide theoretical analysis of the final estimator $\hat{\boldsymbol{\theta}}_T$ under the setting where $\varepsilon = 0$, i.e., all tasks share a common latent parameter within a bounded deviation. The main result is presented in Theorem 1, where we use a unified regularization parameter $\lambda$ to denote $\lambda_t$ and $\lambda_{\mathrm{past}}$ for notational simplicity. To derive the performance guarantee, we impose the following assumptions.

**Assumption 2.** *For any $t \in [T]$ and $z_t \sim \mathcal{D}_t$, $\ell_t(\boldsymbol{\omega}, z_t) : \mathbb{R}^p \to \mathbb{R}$ is convex and twice differentiable. There exist constants $c_1, c_2 > 0$ and $c_1 < \rho, \tau, R < c_2$ such that $\rho \boldsymbol{I} \preceq \nabla^2 \mathcal{L}_t(\boldsymbol{\omega}) \preceq \tau \boldsymbol{I}$ holds for all $\boldsymbol{\omega} \in \mathcal{B}(\boldsymbol{\omega}_t^*, R)$.*

**Assumption 3.** *There exist $0 \leq \sigma, \zeta < c_3$ such that for any $t \in [T]$, the gradient of the empirical loss is $\sigma$-sub-Gaussian. The Hessian matrix, evaluated on a unit vector, is $\zeta$-sub-exponential. Namely,*

$$\|\nabla \ell_t(\boldsymbol{\omega}_t^*, z_t)\|_{\psi_2} \leq \sigma,$$

$$\|\langle \boldsymbol{\xi}, (\nabla^2 \ell_t(\boldsymbol{\omega}, z_t) - \mathbb{E}[\nabla^2 \ell_t(\boldsymbol{\omega}, z_t)]) \boldsymbol{\xi}\rangle\|_{\psi_1} \leq \zeta, \quad \forall \boldsymbol{\omega} \in \mathcal{B}(\boldsymbol{\omega}_t^*, R), \boldsymbol{\xi} \in \mathbb{S}^{p-1}$$

*Further, the Hessian of the loss function is Lipschitz continuous with integrable Lipschitz constant. Namely, there exist a constant $c_4$ such that*

$$\mathbb{E}[H_t(z_t)] \leq \zeta^3 p^{c_4},$$

*where we define*

$$H_t(z_t) = \sup_{\substack{\boldsymbol{\omega}_1, \boldsymbol{\omega}_2 \in \mathcal{B}(\boldsymbol{\omega}_t^*, R) \\ \boldsymbol{\omega}_1 \neq \boldsymbol{\omega}_2}} \frac{\|\nabla^2 \ell_t(\boldsymbol{\omega}_2, z_t) - \nabla^2 \ell_t(\boldsymbol{\omega}_1, z_t)\|_2}{\|\boldsymbol{\omega}_2 - \boldsymbol{\omega}_1\|_2}, \quad \forall z_t \sim \mathcal{D}_t.$$

Assumption 2 requires that the Hessian matrix of the population loss function $\mathcal{L}_t$ for each task $t \in [T]$ is uniformly bounded from above and below in a neighborhood around the true parameter $\boldsymbol{\omega}_t^*$. Note that this assumption only enforces local smoothness around each $\boldsymbol{\omega}_t^*$, but places no restrictions on how $\boldsymbol{\omega}_t^*$ shifts, and thus does not limit the impact of concept drift. Assumption 3 imposes light-tailedness and smoothness conditions on the empirical gradient and Hessian. While these assumptions may be restrictive for non-smooth models such as deep networks, they are broadly used in statistical machine learning, see Mei et al. (2018); Duan and Wang (2023).

**Theorem 1** (Overall performance). *Suppose Assumptions 1–3 hold and that the tasks are $(\varepsilon, \delta)$-related with $\varepsilon = 0$. Then, for positive constants $\{C_i\}_{i=0}^6$, under the scaling conditions $n > C_1 p \log n \log T$, $0 \le \alpha < C_2 n/(p \log n)$, and $C_3 \sigma \sqrt{(p + \log T + \alpha)/n} + C_4 \sigma \sqrt{(p + \alpha)/M} + C_5 \sigma \sqrt{(p + \alpha)/n(T - 1)} < \lambda < C_6 \sigma$, we have the following bound with probability at least $1 - e^{-\alpha}$,*

$$\frac{1}{T} \sum_{t=1}^T \|\hat{\boldsymbol{\theta}}_T - \boldsymbol{\omega}_t^*\|_2 \le C_0 \left( \sigma \frac{\sqrt{p + \alpha}}{n + M} \left\{ \sqrt{n + \frac{M^2}{n(T - 1)}} + \sqrt{M} \right\} + \min\{\delta, \lambda\} \right).$$

*Furthermore, if $\lambda$ is sufficiently large, we have $\hat{\boldsymbol{\theta}}_T = \arg\min_{\boldsymbol{\omega} \in \mathbb{R}^p} \{ML_{past}(\boldsymbol{\omega}) + nL_T(\boldsymbol{\omega})\}/(n + M)$.*

Theorem 1 provides a high-probability bound on the overall performance of $\hat{\boldsymbol{\theta}}_T$. Several important implications follow:

- *Task Similarity*: When the true task parameters $\boldsymbol{\omega}_t^*$ are similar (i.e., $\delta$ is small), knowledge transfer across tasks is beneficial. In this case, a large regularization parameter $\lambda$ is preferred, and the proposed problem essentially reduces to a memory-based method.

- *Regularization Tuning*: In the presence of concept drift (i.e., non-negligible $\delta$), the regularization parameter $\lambda$ should be chosen to balance model plasticity and stability. The theorem suggests setting $\lambda$ on the order of $\lambda \asymp \sqrt{(p + \log T)/n} + \sqrt{p/M} + \sqrt{p/n(T - 1)}$, where the constant can be selected adaptively using the tuning procedure described in Section 3.2.

- *Memory Buffer Size*: The estimation error decreases as the memory buffer size increases. In the extreme case where $M = n(T - 1)$ and all past data are stored, the procedure reduces to the oracle estimator that minimizes the full average population loss equation 1, achieving the optimal rate $\sqrt{p/(nT)}$ when all true task models are identical. In practice, however, storing all historical data is computationally infeasible. The proposed framework mitigates this by using a compact memory buffer of size $M \ll n(T - 1)$, whose efficacy has been empirically validated in prior studies such as Chaudhry et al. (2019).

## 5 EXPERIMENTS

In this section, we evaluate the empirical performance of the proposed method using both synthetic datasets and a real-world application. For the synthetic experiments, results are averaged over 100 independent replications. Unlike benchmarks such as Permuted MNIST or Split CIFAR-10, which test forgetting but do not represent real concept drift, our synthetic settings are designed to match the task similarity structure assumed in our theory. We compare our approach against several continual learning baselines, including fine-tuning via stochastic gradient descent (SGD), elastic weight consolidation (EWC), experience replay (ER), and average gradient episodic memory (AGEM). To ensure a comprehensive comparison, we first outline the evaluation metrics as follows:

**Evaluation** Let $\Delta_t(\boldsymbol{\omega})$ denote the performance of model $\boldsymbol{\omega}$ on task $t$. For instance, in regression problems, $\Delta_t(\boldsymbol{\omega})$ may represent the mean squared error, whereas in classification settings, it may refer to classification accuracy. In all cases, performance is evaluated on a held-out test set specific to each task. We evaluate the proposed method using two standard CL metrics: overall generalization and average forgetting. The overall performance of the final model $\hat{\boldsymbol{\theta}}_T$ is measured as $\text{GE} = \sum_{t=1}^T \Delta_t(\hat{\boldsymbol{\theta}}_T)/T$, reflecting its balance between stability and plasticity. Average forgetting quantifies the performance loss on earlier tasks after sequential updates. Its definition depends on whether the performance metric $\Delta_t(\cdot)$ represents an error or an accuracy measure: If $\Delta_t(\cdot)$

measures error, it is $\mathrm{FE} = \sum_{t=1}^{T-1}\{\Delta_t(\hat{\boldsymbol{\theta}}_T) - \Delta_t(\hat{\boldsymbol{\theta}}_t)\}/(T-1)$. If $\Delta_t(\cdot)$ measures accuracy, it is $\mathrm{FE} = \sum_{t=1}^{T-1}\{\Delta_t(\hat{\boldsymbol{\theta}}_t) - \Delta_t(\hat{\boldsymbol{\theta}}_T)\}/(T-1)$.

**Example 5.1** (**Low-dimensional Synthetic Data**). We consider a setting with $T = 20$ tasks. For each task $t \in [T]$, we set the task size to $n = 2500$ and $p = 50$. Each task generates data $z_t^i = (\mathbf{x}_t^i, y_t^i)$ for $i \in [n]$, where $\mathbf{x}_t^i$ are sampled from $\mathcal{N}(0, \boldsymbol{I})$ and $y_t^i$ the response generated follows a linear model $y_t^i = (\mathbf{x}_t^i)^\top \boldsymbol{\omega}_t^* + e_t^i$ with $e_t^i \sim \mathcal{N}(0, 0.25)$. Task similarity is controlled by parameters $\varepsilon$ and $\delta$, in Assumption 1. To generate the true task-specific coefficients $\{\boldsymbol{\omega}_t^*\}_{t=1}^T$ we proceed as follows: first set a common center $\boldsymbol{\omega}_0 = 2\boldsymbol{e}_1$, and then draw i.i.d. perturbations $\boldsymbol{\delta}_t$ uniformly from the sphere $\delta\mathbb{S}^{p-1}$. Each task coefficient is initialized as $\boldsymbol{\omega}_t^* = \boldsymbol{\omega}_0 + \boldsymbol{\delta}_t$. To introduce heterogeneity, we randomly select $\lceil \varepsilon T \rceil$ task indices and overwrite their $\boldsymbol{\omega}_t^*$ with i.i.d. vectors drawn from $2\mathbb{S}^{p-1}$.

For benchmarking, we construct an offline oracle estimator by minimizing the aggregated population loss in equation 1, assuming access to all task data. For our proposed method, the regularization parameters $\lambda_{\mathrm{past}}$ and $\lambda_t$ follow the theoretical scaling $\sqrt{(p + \log t)/n} + \sqrt{p/m} + \sqrt{p/n(t-1)}$, as suggested by Theorem 1. Candidate constants are chosen from $\{0.01, 0.1, 1, 100000\}$ via the adaptive tuning procedure in Algorithm 2, where the largest value allows the method to mimic ER. The memory buffer size is fixed at $M = 300$. The learning rate for SGD is set to $0.001$, aligning with recommendations from Ding et al. (2024) that smaller step sizes improve generalization performance and algorithmic stability. The ER baseline uses the same learning rate and buffer size. The EWC regularization parameter is set to $0.01$. AGEM is conducted with a mini-batch size of 32 and trained for 5 epochs per task using SGD with step size $0.01$.

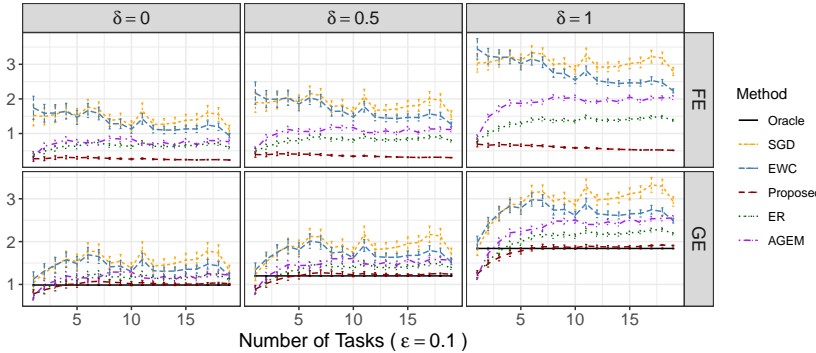

Figure 1: Comparison of generalization and forgetting errors across different task relatedness for the linear case ($\varepsilon = 0.1$), with error bars representing standard error.

We vary $\varepsilon$ in $\{0, 0.1, 0.2\}$ and $\delta$ in $\{0, 0.5, 1\}$ to simulate different levels of task relatedness. Figure 1 reports the GE and FE for $\varepsilon = 0.1$, where error bars denote the standard error across 100 Monte Carlo replicates. As more tasks are processed, GE increases due to the accumulation of heterogeneity. Compared to fine-tuning with SGD, ER and AGEM improve stability by leveraging past samples. Nevertheless, our proposed method, by jointly regularizing past and current objectives, achieves the lowest GE and FE and approaches the oracle performance over time. We also observe that EWC and SGD exhibit sharp fluctuations, indicating sensitivity to large distributional shifts across tasks. In contrast, our method yields consistently more robust performance, especially under higher levels of heterogeneity. Results for $\varepsilon = 0, 0.2$ in Appendix A.2 further support these findings.

**Example 5.2** (**High-dimensional Synthetic Data**). We consider a high-dimensional classification problem solved using support vector machines. Details of the data generation process and model setup are provided in Appendix A.2. Table 1 reports the average classification accuracy across different levels of task relatedness, along with standard errors in parentheses. The results demonstrate that the proposed method consistently achieves superior performance relative to the baselines. Although the theoretical assumptions from Section 4 are not strictly satisfied in this empirical setting, the method remains robust and performs competitively. Figure 2 presents the total runtime for all methods. As expected, the offline oracle estimator incurs the greatest cost, while online methods like EWC and AGEM also demand substantial computation due to multiple training epochs per task. Our

method incurs a slightly higher cost than ER due to the additional overhead associated with dynamic parameter tuning, yet remains computationally efficient in practice.

Table 1: Comparison of average accuracy (standard error $\times 10^{-3}$ in parentheses) across different task relatedness for the high-dimensional classification problem.

| | | | | Method | | | |
|---|---|---|---|---|---|---|---|
| $\varepsilon$ | $\delta$ | SGD | EWC | **Proposed** | ER | AGEM | Oracle |
| 0 | 0 | $0.905_{(0.382)}$ | $0.927_{(0.789)}$ | $0.941_{(0.303)}$ | $0.911_{(0.340)}$ | $0.937_{(0.246)}$ | $0.960_{(0.205)}$ |
| | 0.5 | $0.895_{(0.378)}$ | $0.916_{(0.739)}$ | $0.929_{(0.258)}$ | $0.902_{(0.311)}$ | $0.924_{(0.257)}$ | $0.946_{(0.246)}$ |
| | 1 | $0.873_{(0.407)}$ | $0.892_{(0.713)}$ | $0.903_{(0.289)}$ | $0.879_{(0.328)}$ | $0.897_{(0.331)}$ | $0.918_{(0.285)}$ |
| 0.1 | 0 | $0.835_{(3.210)}$ | $0.856_{(2.500)}$ | $0.872_{(1.077)}$ | $0.839_{(1.804)}$ | $0.865_{(2.680)}$ | $0.902_{(0.294)}$ |
| | 0.5 | $0.829_{(3.069)}$ | $0.849_{(2.413)}$ | $0.865_{(0.990)}$ | $0.832_{(1.736)}$ | $0.858_{(2.460)}$ | $0.893_{(0.257)}$ |
| | 1 | $0.811_{(2.858)}$ | $0.832_{(1.948)}$ | $0.845_{(0.892)}$ | $0.816_{(1.559)}$ | $0.836_{(2.152)}$ | $0.870_{(0.312)}$ |
| 0.2 | 0 | $0.773_{(3.882)}$ | $0.796_{(2.752)}$ | $0.808_{(1.557)}$ | $0.776_{(2.093)}$ | $0.795_{(4.222)}$ | $0.846_{(0.349)}$ |
| | 0.5 | $0.769_{(3.730)}$ | $0.790_{(2.670)}$ | $0.802_{(1.424)}$ | $0.772_{(2.056)}$ | $0.789_{(4.021)}$ | $0.839_{(0.360)}$ |
| | 1 | $0.756_{(3.462)}$ | $0.778_{(2.424)}$ | $0.788_{(1.219)}$ | $0.760_{(1.901)}$ | $0.775_{(3.689)}$ | $0.822_{(0.365)}$ |

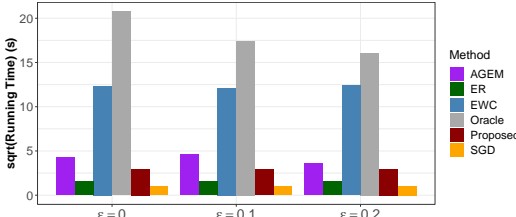

Figure 2: Comparison of total running time (on a square root scale) for the classification problem.

**Example 5.3** (**Real Data on Kidney Transplantation**). We evaluate the proposed method on a real-world dataset from the Organ Procurement and Transplantation Network (`https://optn.transplant.hrsa.gov/`), comprising 359,480 kidney transplant recipients across 200 centers (337 to 6,357 patients per center). The outcome is failure time, measured in days from transplantation to graft failure or death. Ten covariates are used for risk prediction, spanning patient (age, race, gender, BMI, kidney status, insulin use), donor (cold ischemic time, donor status), and organ transport (preservation method, distance) factors. Following Mo et al. (2024), each center is treated as a separate task to capture heterogeneity in regression coefficients.

This problem can be framed as a CL task with linear regression due to its large sample size and heterogeneity across centers. For each task, we randomly select 20% of the data as a held-out test set and train linear models on the remaining 80%. Due to the high skewness of the failure time distribution, we apply a Box-Cox transformation to the response variable, as recommended in Mo et al. (2024). We compare the proposed method against several benchmark approaches: SGD, EWC, ER, AGEM, and an offline oracle estimator that pools data across tasks (Oracle), using standardized data. For the proposed method, regularization parameters are scaled following the same scheme as in the synthetic datasets, with constants selected from the candidate set $\{0.05, 0.1, 0.15, 0.2, 1, 100000\}$. The memory buffer size is fixed at $M = 1000$.

Table 2: Test error on the kidney transplantation dataset.

| | Proposed | SGD | EWC | ER | AGEM | Oracle |
|---|---|---|---|---|---|---|
| GE | **0.955** | 0.995 | 0.973 | 0.967 | 0.979 | 0.952 |
| FE | **0.012** | 0.075 | 0.075 | 0.043 | 0.079 | – |

To evaluate performance, we compute the GE and FE on the test sets using the final model obtained after all tasks have been processed. As shown in Table 2, the proposed method achieves the best generalization performance while also exhibiting superior retention of knowledge from earlier tasks. In contrast, SGD suffers from severe forgetting, highlighting the necessity of mechanisms that address stability-plasticity trade-offs in CL.

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

# A APPENDIX

This appendix presents the memory update algorithm, supplementary numerical results, a discussion of limitations, and the technical proofs of Theorem 1.

## A.1 MEMORY UPDATE ALGORITHM

---
**Algorithm 3** Reservoir Sampling

---
1: **Input:** Data point $z_t^i$, memory buffer $\mathcal{M}^{i-1}$, count, buffer size $M$
2: Count the number of samples currently stored in the memory $m \leftarrow |\mathcal{M}^{i-1}|$
3: count $\leftarrow$ count + 1
4: **if** $m < M$ **then**
5:     Append the memory buffer with the new data point $\mathcal{M}^i \leftarrow c(\mathcal{M}^{i-1}, z_t^i)$
6: **else**
7:     Generate a random number $j$ uniformly in $\{1, \dots, \text{count}\}$
8:     **if** $j < M$ **then**
9:         Overwrite memory slot $\mathcal{M}_j \leftarrow z_t^i$
10:     **end if**
11: **end if**
12: **Output:** $\mathcal{M}^i$, count

---

## A.2 ADDITIONAL NUMERICAL RESULTS

**Example A.1** (**High-dimensional Synthetic Data**). We consider a high-dimensional classification problem with $T = 20$ tasks, each consisting of $n = 2000$ and $p = 1500$. The covariate vectors are generated independently from a standard multivariate normal distribution, $\mathbf{x}_t^i \sim \mathcal{N}(0, \boldsymbol{I})$. The reponse variable is simulated according to a probit model: $\mathbb{P}(y_t^i = 1 \mid \mathbf{x}_t^i) = \Phi\{(\mathbf{x}_t^i)^\top \boldsymbol{\omega}_t^*\}$, where $\Phi(\cdot)$ is the cumulative density function of the standard normal distribution. The task-specific true parameter vectors $\{\boldsymbol{\omega}_t^*\}_{t=1}^T$ are generated using the same procedure as in the low-dimensional case, with the central parameter fixed as $\boldsymbol{\omega}_0 = (2, 2, 2, 2, 2, 0, \ldots, 0)^\top$. We employ a support vector machine with a linear kernel and regularization parameter set to $0.1$.

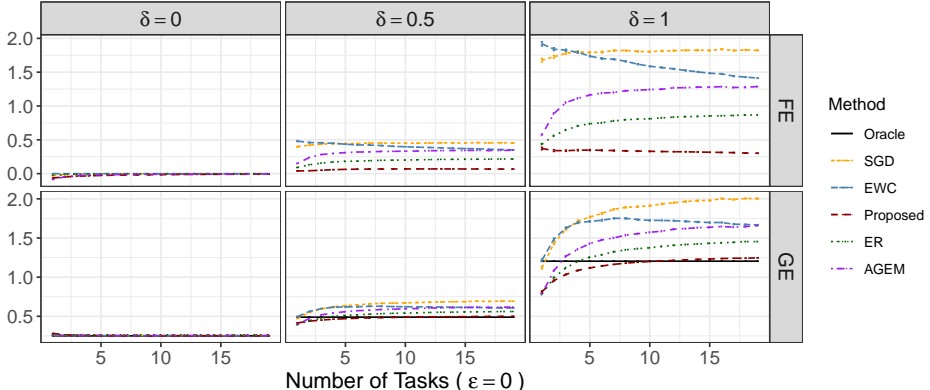

Figure 3: Comparison of generalization and forgetting errors across different task relatedness for the linear case ($\varepsilon = 0$), with error bars representing standard error.

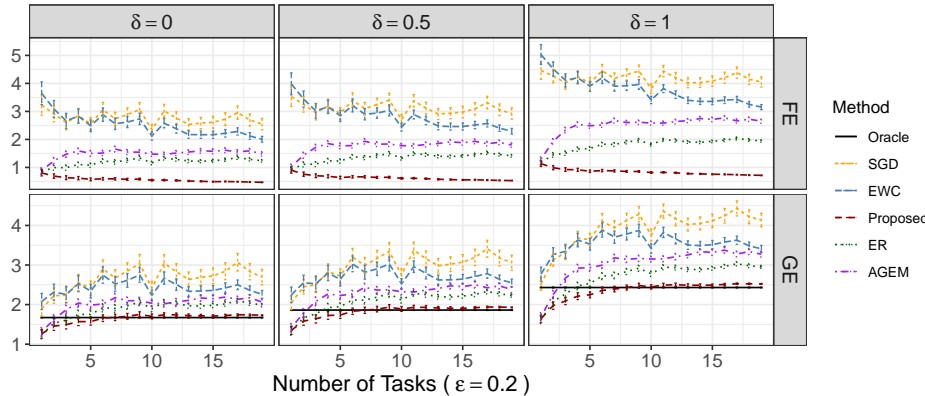

Figure 4: Comparison of generalization and forgetting errors across different task relatedness for the linear case ($\varepsilon = 0.2$), with error bars representing standard error.

## A.3 LIMITATION

Our current framework assumes known task boundaries. While effective, this approach could be limiting in more general continual learning settings, such as task-free scenarios. Reservoir sampling offers the advantage of not requiring task boundary information for memory management. Extending this flexibility to our framework would broaden its applicability. Additionally, for imbalanced task sequences, replacing reservoir sampling with a gradient-based sampling method could improve the representation of tasks that are underrepresented in the distribution (Aljundi et al., 2019b). We leave both extensions as directions for future work.

A.4 TECHNICAL PROOFS

Throughout the proofs, we will flexibly use bounded constants such as $c_1, c_2, C_1, C_2$, which may differ from line to line. We denote the infimal convolution of two convex functions $f$ and $g$ by $f \square g$, defined as $f \square g(\boldsymbol{\nu}) = \inf_{\boldsymbol{\omega} \in \mathbb{R}^p} \{ f(\boldsymbol{\omega}) + g(\boldsymbol{\omega} - \boldsymbol{\nu}) \}$.

**Lemma 1.** *Let Assumption 2 hold and $\|\nabla L_t(\boldsymbol{\omega}_t^*)\|_2 \leq \rho R / 2$. If $0 < \lambda_t < \rho R / 2$, we have*

$$\|\hat{\boldsymbol{\theta}} - \boldsymbol{\omega}_t^*\|_2 \leq \frac{\|\nabla L_t(\boldsymbol{\omega}_t^*)\|_2}{\rho} + \frac{\lambda_t}{\rho}.$$

*Proof of Lemma 1.* The proof can be found in Theorem A.1 of Duan and Wang (2023). □

**Lemma 2.** *Let $\{f_t\}_{t=1}^2$ be convex and differentiable. Define $F = \sum_{t=1}^2 a_t f_t \square (\lambda_t \| \cdot \|_2)$ and $G = \sum_{t=1}^2 a_t f_t$. Suppose there exist $\boldsymbol{\omega}^* \in \mathbb{R}^p, 0 < R \leq \infty$ and $0 < \rho_0, \tau_1, \tau_2 < \infty$ such that*

$$\nabla^2 f_t(\boldsymbol{\omega}) \preceq \tau_t \boldsymbol{I}, \quad t = 1, 2 \quad and \quad \nabla^2 G(\boldsymbol{\omega}) \succeq \rho_0 \boldsymbol{I}$$

*hold for all $\boldsymbol{\omega} \in \mathcal{B}(\boldsymbol{\omega}^*, R)$. If*

$$\|\nabla f_t(\boldsymbol{\omega}^*)\|_2 + \frac{2\tau_t \left\| \sum_{k=1}^2 a_k \nabla f_k(\boldsymbol{\omega}^*) \right\|_2}{\rho_0} < \lambda_t < \|\nabla f_t(\boldsymbol{\omega}^*)\|_2 + \tau_t R, \quad for \ t = 1, 2,$$

*then $\hat{\boldsymbol{\omega}}_1 = \hat{\boldsymbol{\omega}}_2 = \hat{\boldsymbol{\theta}} = \tilde{\boldsymbol{\omega}}$, where $\tilde{\boldsymbol{\omega}} \in \arg\min_{\boldsymbol{\omega}} \{ \sum_{t=1}^2 a_t f_t(\boldsymbol{\omega}) \}$,*

$$F(\boldsymbol{\omega}) = G(\boldsymbol{\omega}), \quad \forall \boldsymbol{\omega} \in \mathcal{B}(\boldsymbol{\omega}^*, \min \{ (\lambda_t - \|\nabla f_t(\boldsymbol{\omega}^*)\|_2) / \tau_t \}).$$

*We also have*

$$\left\| \hat{\boldsymbol{\theta}} - \boldsymbol{\omega}^* \right\|_2 \leq \frac{\left\| \sum_{t=1}^2 a_t \nabla f_t(\boldsymbol{\omega}^*) \right\|_2}{\rho_0}.$$

*Proof of Lemma 2.* The proof can be found in Lemma B.2 of Duan and Wang (2023). □

**Lemma 3.** *Let Assumptions 2 and 3 hold. There exist constants $C$, $C_1$ and $C_2$ such that under the conditions $n > C_1 p \log n \log T$ and $0 \leq \alpha < C_2 n / (p \log n)$, the following hold with probability at least $1 - e^{-\alpha}$:*

$$\|\nabla L_t(\boldsymbol{\omega}_t^*)\|_2 < C\sigma \sqrt{\frac{p + \log T + \alpha}{n}} \leq \frac{\rho R}{4}, \quad \forall t \in [T];$$

$$\frac{\rho}{2} \boldsymbol{I} \preceq \nabla^2 L_t(\boldsymbol{\omega}) \preceq \frac{3\tau}{2} \boldsymbol{I}, \quad \forall \boldsymbol{\omega} \in \mathcal{B}(\boldsymbol{\omega}_t^*, R), \quad t \in [T].$$

*Proof of Lemma 3.* The proof can be found in Lemma D.1 of Duan and Wang (2023). □

*Proof of Theorem 1.* By Lemma 3, $\frac{\rho}{2} \boldsymbol{I} \preceq \nabla^2 L_t(\boldsymbol{\omega}) \preceq \frac{3\tau}{2} \boldsymbol{I}$ for $\forall \boldsymbol{\omega} \in \mathcal{B}(\boldsymbol{\omega}_t^*, R)$. Define

$$\boldsymbol{\omega}^* = \operatorname{argmin}_{\boldsymbol{\omega} \in \mathbb{R}^p} \max_{t \in [T]} \|\boldsymbol{\omega}_t^* - \boldsymbol{\omega}\|_2.$$

Under Assumption 1, the regularity condition $\nabla^2 L_t(\boldsymbol{\omega}) \preceq \tau \boldsymbol{I}$ for $\forall \boldsymbol{\omega} \in \mathcal{B}(\boldsymbol{\omega}_t^*, R)$ leads to $\|\nabla L_t(\boldsymbol{\omega}_t^*) - \nabla L_t(\boldsymbol{\omega}^*)\|_2 \leq \tau \delta$. By triangle inequality, $\|\nabla L_t(\boldsymbol{\omega}^*)\|_2 \leq \|\nabla L_t(\boldsymbol{\omega}_t^*)\|_2 + \tau \delta$ and

$$\|\nabla L_{\text{past}}(\boldsymbol{\omega}^*)\|_2 \leq \| \frac{1}{t-1} \sum_{k=1}^{t-1} \nabla L_k(\boldsymbol{\omega}_k^*) \|_2 + \frac{1}{t-1} \| \sum_{k=1}^{t-1} \nabla L_k(\boldsymbol{\omega}^*) - \sum_{k=1}^{t-1} \nabla L_k(\boldsymbol{\omega}_k^*) \|_2$$

$$+ \|\nabla L_{\text{past}}(\boldsymbol{\omega}^*) - \frac{1}{t-1} \sum_{k=1}^{t-1} \nabla L_k(\boldsymbol{\omega}^*) \|_2$$

$$\leq \| \frac{1}{t-1} \sum_{k=1}^{t-1} \nabla L_k(\boldsymbol{\omega}_k^*) \|_2 + \tau \delta + \Delta_{\text{past}},$$

where $\Delta_{\text{past}} := \|\nabla L_{\text{past}}(\boldsymbol{\omega}^*) - \frac{1}{t-1}\sum_{k=1}^{t-1}\nabla L_k(\boldsymbol{\omega}^*)\|_2$. To find the upper bound of $\Delta_{\text{past}}$, we use triangle inequality:

$$\Delta_{\text{past}} \leq \|\frac{1}{m}\sum_{(k,i)\in\mathcal{M}}\nabla\ell_{k,i}(\boldsymbol{\omega}_k^*) - \frac{1}{t-1}\sum_{k=1}^{t-1}\frac{1}{n}\sum_{i=1}^{n}\nabla\ell_{k,i}(\boldsymbol{\omega}_k^*)\|_2 + \|\frac{1}{m}\sum_{(k,i)\in\mathcal{M}}\{\nabla\ell_{k,i}(\boldsymbol{\omega}^*) - \nabla\ell_{k,i}(\boldsymbol{\omega}_k^*)\}\|_2$$

$$+ \|\frac{1}{n(t-1)}\sum_{k=1}^{t-1}\sum_{i=1}^{n}\{\nabla\ell_{k,i}(\boldsymbol{\omega}^*) - \nabla\ell_{k,i}(\boldsymbol{\omega}_k^*)\}\|_2$$

$$\lesssim \|\frac{1}{m}\sum_{(k,i)\in\mathcal{M}}\nabla\ell_{k,i}(\boldsymbol{\omega}_k^*)\|_2 + \|\frac{1}{t-1}\sum_{k=1}^{t-1}\frac{1}{n}\sum_{i=1}^{n}\nabla\ell_{k,i}(\boldsymbol{\omega}_k^*)\|_2 + \tau\delta.$$

As $\frac{1}{m}\sum_{(k,i)\in\mathcal{M}}\nabla\ell_{k,i}(\boldsymbol{\omega}_k^*)$ and $\frac{1}{t-1}\sum_{k=1}^{t-1}\frac{1}{n}\sum_{i=1}^{n}\nabla\ell_{k,i}(\boldsymbol{\omega}_k^*)$ have zero means, and

$$\|\frac{1}{m}\sum_{(k,i)\in\mathcal{M}}\nabla\ell_{k,i}(\boldsymbol{\omega}_k^*)\|_{\psi_2} \lesssim \frac{\sigma}{\sqrt{m}}, \quad \|\frac{1}{t-1}\sum_{k=1}^{t-1}\frac{1}{n}\sum_{i=1}^{n}\nabla\ell_{k,i}(\boldsymbol{\omega}_k^*)\|_{\psi_2} \lesssim \frac{\sigma}{\sqrt{n(t-1)}}.$$

By Theorem 2.1 in Hsu et al. (2012), for all $\alpha > 0$, we can find universal constants $c_1'$ and $c_2'$ such that with probability at least $1 - e^{-\alpha}$,

$$\|\frac{1}{m}\sum_{(k,i)\in\mathcal{M}}\nabla\ell_{k,i}(\boldsymbol{\omega}_k^*)\|_2 + \|\frac{1}{t-1}\sum_{k=1}^{t-1}\frac{1}{n}\sum_{i=1}^{n}\nabla\ell_{k,i}(\boldsymbol{\omega}_k^*)\|_2 \leq c_1'\sigma\sqrt{\frac{p+\alpha}{m}} + c_2'\sigma\sqrt{\frac{p+\alpha}{n(t-1)}}.$$

Hence,

$$\Delta_{\text{past}} \lesssim \sigma\left\{\sqrt{\frac{p+\alpha}{m}} + \sqrt{\frac{p+\alpha}{n(t-1)}}\right\} + \tau\delta. \tag{6}$$

Let $\eta = \max_{t\in[T]}\{\|\nabla L_t(\boldsymbol{\omega}^*)\|_2\}$ and $\kappa = \tau/\rho$, we get

$$\|\nabla L_t(\boldsymbol{\omega}^*)\|_2 + \frac{2\tau\|\frac{m}{n+m}\nabla L_{\text{past}}(\boldsymbol{\omega}^*) + \frac{n}{n+m}\nabla L_t(\boldsymbol{\omega}^*)\|_2}{\rho}$$

$$\leq \eta + \frac{2\tau\{\frac{m}{n+m}\|\frac{1}{t-1}\sum_{k=1}^{t-1}\nabla L_k(\boldsymbol{\omega}^*)\|_2 + \|\frac{n}{n+m}\nabla L_t(\boldsymbol{\omega}^*)\|_2\}}{\rho} + \frac{2\tau m\Delta_{\text{past}}}{(n+m)\rho}$$

$$\lesssim \eta(1 + \frac{2\tau}{\rho}) + \frac{2\tau m}{(n+m)\rho}\left\{\sigma\sqrt{\frac{p+\alpha}{m}} + \sigma\sqrt{\frac{p+\alpha}{n(t-1)}} + \tau\delta\right\}$$

$$\leq 3\kappa\eta + 2\kappa\tau\delta + 2\kappa\sigma\left\{\sqrt{\frac{p+\alpha}{m}} + \sqrt{\frac{p+\alpha}{n(t-1)}}\right\}.$$

Let $g = \max_{t\in[T]}\{\|\nabla L_t(\boldsymbol{\omega}_t^*)\|_2\}$. If we assume $3\tau\delta \leq g + \frac{\lambda_t}{5\kappa}$, by triangle inequality, $\eta \leq g + \tau\delta \leq \frac{4g}{3} + \frac{\lambda_t}{15\kappa}$. Therefore, when $\lambda_t$ satisfies

$$7\kappa g + 3\kappa\sigma\left\{\sqrt{\frac{p+\alpha}{m}} + \sqrt{\frac{p+\alpha}{n(t-1)}}\right\} < \lambda_t < \frac{\rho R}{2},$$

we have

$$3\kappa\eta + 2\kappa\tau\delta + 2\kappa\sigma\left\{\sqrt{\frac{p+\alpha}{m}} + \sqrt{\frac{p+\alpha}{n(t-1)}}\right\}$$

$$< \frac{14}{3}\kappa g + \frac{\lambda_t}{3} + 2\kappa\sigma\left\{\sqrt{\frac{p+\alpha}{m}} + \sqrt{\frac{p+\alpha}{n(t-1)}}\right\}$$

$$< \lambda_t < \frac{\rho R}{2} < \frac{4\tau R}{5},$$

which satisfies the assumption in Lemma 2. Let $f_1 := L_{\text{past}}, f_2 := L_t, a_1 = \frac{m}{n+m}, a_2 = \frac{n}{n+m}$. By Lemma 2 and triangle inequality,

$$\|\hat{\boldsymbol{\theta}} - \boldsymbol{\omega}^*\|_2 \leq \frac{\|\frac{m}{n+m}\nabla L_{\text{past}}(\boldsymbol{\omega}^*) + \frac{n}{n+m}\nabla L_t(\boldsymbol{\omega}^*)\|_2}{\rho}$$

$$\leq \frac{\|G\|_2}{\rho} + \frac{\tau\delta}{\rho} + \frac{m\Delta_{\text{past}}}{(n+m)\rho},$$

where $G := \frac{m}{(n+m)(t-1)}\sum_{k=1}^{t-1}\nabla L_k(\boldsymbol{\omega}_k^*) + \frac{n}{n+m}\nabla L_t(\boldsymbol{\omega}_t^*)$. Assumption 1 yields that

$$\|\hat{\boldsymbol{\theta}} - \boldsymbol{\omega}_t^*\|_2 \leq 2\kappa\delta + \frac{m\Delta_{\text{past}}}{(n+m)\rho} + \frac{\|G\|_2}{\rho}.$$

As $3\tau\delta \leq g + \frac{\lambda_t}{5\kappa}$, we have

$$\|\hat{\boldsymbol{\theta}} - \boldsymbol{\omega}_t^*\|_2 \leq \frac{1}{\rho}\min\{3\tau\delta, g + \frac{\lambda_t}{5\kappa}\} + \frac{m\Delta_{\text{past}}}{(n+m)\rho} + \frac{\|G\|_2}{\rho}.$$

When $3\tau\delta > g + \frac{\lambda_t}{5\kappa}$, since the regularization terms $\|\cdot\|_2$ in equation 2 are convex, we know $\|\hat{\boldsymbol{\theta}} - \boldsymbol{\omega}_t^*\|_2 \lesssim \max\{\|\hat{\boldsymbol{\omega}}_{\text{past}} - \boldsymbol{\omega}_t^*\|_2, \|\hat{\boldsymbol{\omega}}_t - \boldsymbol{\omega}_t^*\|_2\}$. By Lemma 1,

$$\|\hat{\boldsymbol{\theta}} - \boldsymbol{\omega}_t^*\|_2 \lesssim \frac{g + \lambda_t}{\rho} < (\frac{1}{5\kappa} + 1)\frac{\lambda_t}{\rho} \leq \frac{6\lambda_t}{5\rho},$$

where the second inequality is due to $\lambda_t > 5\kappa g$. In summary,

$$\|\hat{\boldsymbol{\theta}} - \boldsymbol{\omega}_t^*\|_2 \lesssim \frac{\|G\|_2}{\rho} + \frac{m\Delta_{\text{past}}}{(n+m)\rho} + \frac{6\kappa}{\rho}\min\{3\tau\delta, g + \frac{\lambda_t}{5\kappa}\}$$

$$\leq \frac{\|G\|_2}{\rho} + \frac{m\Delta_{\text{past}}}{(n+m)\rho} + \min\{\kappa^2\delta, \frac{\lambda_t}{\rho}\}.$$

Lastly, assuming $\rho, \tau, R \asymp 1$, Lemma 3 allows us to express the condition on $\lambda_t$ as

$$C_1\sigma\sqrt{\frac{p + \log T + \alpha}{n}} + C_2\sigma\sqrt{\frac{p + \alpha}{m}} + C_3\sigma\sqrt{\frac{p + \alpha}{n(t-1)}} < \lambda_t < C_4\sigma,$$

with some positive constants $\{C_i\}_{i=1}^4$. Note that $\mathbb{E}[G] = 0$ and $\|G\|_{\psi_2} \lesssim \frac{\sigma}{n+m}\sqrt{n + \frac{m^2}{n(t-1)}}$. We can find a universal constant $c'$ such that for all $\alpha > 0$,

$$\mathbb{P}\left(\|G\|_2 \geq c'\sigma\frac{\sqrt{p+\alpha}}{n+m}\sqrt{n + \frac{m^2}{n(t-1)}}\right) \leq e^{-\alpha}/3.$$

Hence, combining with equation 6, we have

$$\|\hat{\boldsymbol{\theta}} - \boldsymbol{\omega}_t^*\|_2 \lesssim \sigma\frac{\sqrt{p+\alpha}}{n+m}\sqrt{n + \frac{m^2}{n(t-1)}} + \frac{m}{n+m}\Delta_{\text{past}} + \min\{\delta, \lambda_t\}$$

$$\lesssim \sigma\frac{\sqrt{p+\alpha}}{n+m}\left\{\sqrt{n + \frac{m^2}{n(t-1)}} + \sqrt{m}\right\} + \min\{\delta, \lambda_t\},$$

with probability at least $1 - e^{-\alpha}$. $\qquad\square$

