# OpenReview forum: "Online Continual Learning under Real Concept Drift: A Statistical Perspective"
_ICLR.cc/2026/Conference — Submitted to ICLR 2026_

### Official Review · Reviewer_s2AM · 2025-10-19

**Soundness:** 3
**Presentation:** 3
**Contribution:** 2
**Rating:** 4
**Confidence:** 4

**Summary:**

This paper presents a framework for continual learning in dynamic environments where the relationship between inputs and outputs changes over time. It proposes a regularization-based method that combines memory replay with adaptive tuning to balance knowledge retention and adaptability. The authors develop an efficient online algorithm with theoretical guarantees linking performance to task similarity, memory size, and regularization strength. Experiments on synthetic and real-world benchmarks show that the proposed method outperforms established continual learning baselines.

**Strengths:**

1. This paper is well-written and easy to follow.

2. The proposed concept drift setting is interesting. It may apply to some specific scenarios such as recommendation systems.

3. The proposed method is theoretically-grounded, and achieves strong performance in this particular setting.

**Weaknesses:**

1. The use of memory buffer, although effective, may result in additional storage cost and privacy concerns.

2. The compared methods are mainly traditional continual learning methods (EWC, ER, and AGEM). Is it possible to include more recent methods, such as continual unlearning methods?

3. The experiments are mainly performed with relatively simple datasets, such as MNIST and CIFAR-10. Does the proposed method apply to larger-scale datasets, such as ImageNet (subsets)?

4. Do the theoretical analysis and the proposed method only apply to online continual learning? Is it possible to extend them to other continual learning scenarios (e.g., offline continual learning)?

**Questions:**

My major concerns lie in the comparison baselines and applicability of the proposed method. Please refer to the Weaknesses.

---

> ### Author Response · Authors · 2025-11-29
>
> We thank the reviewer for the careful review and constructive feedback. Below, we address each point in detail.
>
> ---
>
> **Q1: The use of memory buffer, although effective, may result in additional storage cost and privacy concerns.**
>
> **A1:** We thank the reviewer for the comment. The memory buffer sizes M = 300 or 1000 are relatively small compared with the total sample sizes of 50,000 and 359,480. We will include additional experiments to evaluate performance under even more limited memory constraints. As noted in the ER paper [1], experience replay-based methods can substantially reduce forgetting compared with regularisation-based approaches with only a negligible additional computational cost. Users can choose a method based on their computational resources and the desired balance between accuracy and speed.
>
> Regarding privacy, one way to formally quantify privacy protection is to adopt differential privacy, for example, by using differentially private sampling techniques [2-3]. However, privacy protection is currently beyond the scope of this paper.
>
> ---
>
> **Q2: The compared methods are mainly traditional continual learning methods (EWC, ER, and AGEM). Is it possible to include more recent methods, such as continual unlearning methods?**
>
> **A2:**  Thanks for the suggestion. Continual unlearning methods try to make a model forget specific tasks or data points, usually for privacy reasons [4-5]. They focus on reducing performance on the removed data while keeping other knowledge. Our method, on the other hand, deals with real concept drift; the goal is to keep the model stable and adaptive as it learns new tasks. Because the objectives are different, we do not directly compare with continual unlearning methods, but we will mention them in the related work.
>
> ---
>
> **Q3: The experiments are mainly performed with relatively simple datasets, such as MNIST and CIFAR-10. Does the proposed method apply to larger-scale datasets, such as ImageNet (subsets)?**
>
> **A3:** We want to clarify that the simulation studies are not performed using typical benchmarks such as Permuted MNIST or Split CIFAR-10, which mainly test task-incremental, class-incremental, or simple covariate drift scenarios but do not reflect real concept drift. Instead, our synthetic settings are designed to match the task similarity structure assumed in our theoretical analysis.
>
> ---
>
> **Q4: Do the theoretical analysis and the proposed method only apply to online continual learning? Is it possible to extend them to other continual learning scenarios (e.g., offline continual learning)?**
>
> **A4:** Yes, it is possible with some modifications. For instance, if the proposed method is applied in a traditional batch continual learning setting, the stochastic proximal gradient descent and stochastic gradient descent in Algorithm 1 can be performed on batches of data rather than updating with each new data point. This change does not affect the theoretical analysis. However, offline continual learning can significantly increase computational cost and runtime, as illustrated in Figure 2, where the offline oracle estimator incurs the highest cost.
>
> ---
>
> [1] Chaudhry, A., Rohrbach, M., Elhoseiny, M., Ajanthan, T., Dokania, P., Torr, P. and Ranzato, M., 2019, June. Continual learning with tiny episodic memories. In Workshop on Multi-Task and Lifelong Reinforcement Learning.
>
> [2] Raskhodnikova, S., Sivakumar, S., Smith, A. and Swanberg, M., 2021. Differentially private sampling from distributions. Advances in Neural Information Processing Systems, 34, pp.28983-28994.
>
> [3] Cohen, E., Geri, O., Sarlos, T. and Stemmer, U., 2021, March. Differentially private weighted sampling. In International Conference on Artificial Intelligence and Statistics (pp. 2404-2412). PMLR.
>
> [4] Shibata, T., Irie, G., Ikami, D. and Mitsuzumi, Y., 2021, August. Learning with Selective Forgetting. In IJCAI (Vol. 3, p. 4).
>
> [5] Liu, B., Liu, Q. and Stone, P., 2022, November. Continual learning and private unlearning. In Conference on Lifelong Learning Agents (pp. 243-254). PMLR.

---

### Official Review · Reviewer_YpUb · 2025-10-26

**Soundness:** 2
**Presentation:** 2
**Contribution:** 2
**Rating:** 2
**Confidence:** 4

**Summary:**

The paper introduces a regularization-based method leveraging a memory buffer to address real concept drift. This method connects past and present task estimations through a central point for evolving models, promoting both stability and adaptation. Theoretical bounds regarding generalization error are provided, and their method is benchmarked against SGD, EWC, ER, and AGEM on both synthetic and real-world datasets, including kidney transplantation data.

**Strengths:**

1. **Theoretical Contribution:** The paper contributes to the theoretical understanding of Continual Learning with the buffer size used.

2. **Clarity:** The overall writing is ok and accessible.

**Weaknesses:**

1. **Notation**  As $|\cdot|$ is defined as the absolute value of a real number or cardinality of a set, the definition in Assumption 1 is problematic, which should be  $\|\|w_j -w_0\|\|_2$. In addition, $J^c$ is undefined. In line 179, the definition of empirical loss of the $t$-th task $\ell(w, z_t^i)$ is inconsistent with that in eq.(1), which is for the whole data set.
2. **Insufficient Novelty and Related Work Discussion** I do not find obvious significance for the proposed methods against online/continual meta-learning approaches, where hyperparameters and regularizations are both considered for concept drift or more general shifts. This work is missing related works in this domain, e.g., [1-8].

3. **Theoretical Results:**
    * Theorem 1 only measures the distance of a single $\hat{\theta}_T$ to every task optimal parameter; it's unclear how it affects the average population loss.
    * Unclear definition of the $\lambda$ in Theorem 1, whether all the tasks use the same $\lambda$? What is its relation to the $\lambda$s in eq(2)?
    * How does $a_1$ and $a_2$ affect the theoretical results?

4.  **Empirical Results:**
    * Potential Overfitting in Real-world Data: The paper indicates that theoretical assumptions might not fully apply in empirical evaluations, suggesting possible overfitting or adaptation challenges in diverse settings.
    * The memory buffer size effectively affects the results. How can we make sure it's fairly compared with methods without a memory buffer?
    * Except for Permuted MNIST or Split CIFAR-10, typical online/continual meta-learning approaches consider concept drifts by predicting different characters for each task, e.g., select different five characters from 10 to predict their classes.

---
[1] Chelsea Finn, Aravind Rajeswaran, Sham Kakade, and Sergey Levine. Online meta-learning.
In International Conference on Machine Learning, pages 1920–1930. PMLR, 2019.

[2] Massimo Caccia, Pau Rodriguez, Oleksiy Ostapenko, Fabrice Normandin, Min Lin, Lucas
Page-Caccia, Issam Hadj Laradji, Irina Rish, Alexandre Lacoste, David Vázquez, et al. Online
fast adaptation and knowledge accumulation (osaka): a new approach to continual learning.
Advances in Neural Information Processing Systems, 33:16532–16545, 2020.

[3] Giulia Denevi, Carlo Ciliberto, Riccardo Grazzi, and Massimiliano Pontil. Learning-to-learn
stochastic gradient descent with biased regularization. In International Conference on Machine
Learning, pages 1566–1575. PMLR, 2019.

[4] Qi Chen, Changjian Shui, Ligong Han, and Mario Marchand. On the stability-plasticity dilemma
in continual meta-learning: Theory and algorithm. Advances in Neural Information Processing
Systems, 36:27414–27468, 2023.

[5] Maria-Florina Balcan, Mikhail Khodak, and Ameet Talwalkar. Provable guarantees for gradientbased meta-learning. In International Conference on Machine Learning, pages 424–433. PMLR,
2019.

[6] Mikhail Khodak, Maria-Florina Balcan, and Ameet Talwalkar. Adaptive gradient-based metalearning methods. arXiv preprint arXiv:1906.02717, 2019.

[7] Qiang Zhang, Jinyuan Fang, Zaiqiao Meng, Shangsong Liang, and Emine Yilmaz. Variational
continual bayesian meta-learning. Advances in Neural Information Processing Systems, 34:
24556–24568, 2021.

[8] Xu, Kunlun, et al. "Componential Prompt-Knowledge Alignment for Domain Incremental Learning." arXiv preprint arXiv:2505.04575 (2025).

**Questions:**

1. **Question on the Empirical Results:** When $\delta=0$, which means all the tasks are the same, can you explain why the proposed method is better than others?

2. **Generalization to Complex Tasks:** How does the method perform with tasks having complex dependencies beyond shared central models?

3. **Memory Buffer Adaptability:** Does the method adapt efficiently under varying memory constraints, significantly beyond tested scenarios?

4. **Implementation Details:** Given the overhead of dynamic parameter tuning, are there strategies to streamline this process without compromising learning efficacy?

---

> ### Author Response · Authors · 2025-11-29
> **Responses (Part 1/2)**
>
> We thank the reviewer for the careful review and constructive feedback. Below, we address each point in detail.
>
> ---
>
> **Q1: Notation**
>
> **A1:** Thank you for pointing this out. We have revised the notation in Assumption 1, replacing $|\cdot|$ with the Euclidean norm $|| \cdot ||_2$, and explicitly defined $J^c = [T] \setminus J$. In line 179, we use $\ell_k(\omega, z_t^i)$ to denote the task-specific loss on a single data point $z_t^i$. In eq(1), which describes the population loss, we use $\ell_k(\omega, z_t)$. Although we do not explicitly specify $z_t^i$, since $z_t \sim \mathcal{D}_t$, it empirically represents the individual data points $z_t^i$.
>
> ---
>
> **Q2: Insufficient Novelty and Related Work Discussion**
>
> **A2:** Thank you for providing the literature on online and continual meta-learning. We will expand the related-work section accordingly. We would like to emphasize that our setting differs from (online) meta-learning in an essential way: meta-learning methods aim to enable rapid adaptation to new tasks, whereas our method focuses on retaining task-specific knowledge while adapting to an evolving sequence of tasks. Continual meta-learning is more closely related, as it maintains stability of meta-parameters under task shifts. However, to the best of our knowledge, existing work primarily addresses task-distribution drift or covariate drift, and we are not aware of prior literature that explicitly studies real concept drift, where the conditional relationship between inputs and labels changes while the covariate distribution remains fixed.
> Our main contribution is to formally characterize this concept-drift setting through $(\varepsilon, \delta)$-relatedness and to develop a novel regularization-based method with a memory buffer that directly addresses this scenario. We further provide generalization error guarantees and demonstrate superior empirical performance.
>
> ---
>
> **Q3: Theoretical Results**
>
> **A3:**
>
> * (1) Under Assumption 2, the excess population risk satisfies $\mathcal{L}_t( \hat{\theta}_T ) - \mathcal{L}_t( \omega_t^\star ) \leq (\tau/2) || \hat{\theta}_T - \omega_t^\star ||_2^2$.
> Therefore, Theorem 1 directly implies a bound on the average excess population loss.
>
> * (2)  For notation simplicity, Theorem 1 sets $\lambda := \lambda_{\text{past}} = \lambda_t$. In the algorithm, $\lambda_{\text{past}}$ and $\lambda_t$ are tuned separately, as shown in eq(2), which defines two distinct regularization strengths. The theorem analyzes a symmetric version in order to keep the presentation clear.
>
> * (3)  The coefficients $a_1$ and $a_2$ determine the stability–plasticity trade-off by balancing the influence of replayed gradients versus current-task gradients.  In Theorem 1, we adopt the principled choice $a_1 = {m}/{(n+m)}$ and $a_2 = {n}/{(n+m)}$, so the bound does not explicitly display $a_1$ and $a_2$.
> ---
>
> **Q4: Empirical Reuslts**
>
> **A4:**
>
> * (1) Yes, we agree that Theorem 1 relies on assumptions that may not be fully met in real data. These assumptions are commonly used in statistical machine learning, which we discuss in the paper. In addition, Example 5.2 illustrates the empirical behavior of our method and shows that it consistently delivers strong performance compared with the baselines, even though the theoretical assumptions are not perfectly satisfied in this empirical setting.
>
> * (2)  All replay-based methods (ER, AGEM, and ours) use the same buffer size, and SGD and EWC serve as baselines without a buffer. We state this fairness protocol clearly in the paper. Their total running time is reported in Figure 2. It is up to the user to choose a method based on computational resources and the preferred balance between accuracy and speed.
>
> * (3) Thanks for the suggestion. Although selecting different sets of characters for each task, as in Zhang et al. [1], does introduce a form of non-stationarity, this setting is essentially task incremental learning because each task corresponds to a different classification problem. This is not the situation examined in our work. We study concept drift where the mapping from inputs to labels changes over time while the input distribution stays the same.
>
> ---
>
> [1] Qiang Zhang, Jinyuan Fang, Zaiqiao Meng, Shangsong Liang, and Emine Yilmaz. Variational continual bayesian meta-learning. Advances in Neural Information Processing Systems, 34: 24556–24568, 2021.

---

> ### Author Response · Authors · 2025-11-29
> **Responses (Part 2/2)**
>
> **Q5: Question on the Empirical Results**
>
> **A5:** We would like to clarify that Figure 1 presents the results for $\varepsilon = 0.1$ and $\delta \in \{0, 0.5, 1\}$. When $\delta = 0$, it does not imply that all tasks are identical. In this case, about ten percent of the true task models are still randomly generated. The results in Figure 1 show that our method performs better when this situation occurs. Figure 3 in the Appendix presents the results for $\varepsilon = 0$ and $\delta = 0$, where all tasks share the same underlying model, and in this case all methods behave almost the same.
>
> ---
>
> **Q6: Generalization to Complex Tasks**
>
> **A6:** Thank you for the insightful comment. The performance on the real-world data in Example 5.3 reflects a setting where there may be more complex dependencies than in the simulation study, which assumes a single shared central model. A possible direction for future work is to extend our methodology from a single-center setting to a setting with multiple centers or a low-rank structure. This direction is interesting, but it is beyond the scope of the current paper.
>
> ---
>
> **Q7: Memory Buffer Adaptability**
>
> **A7:** In our experiments, the memory buffer sizes $M = 300$ or $1000$ are relatively small compared with the total sample sizes of $50{,}000$ and $359{,}480$. We will include additional experiments that examine performance under even more limited memory constraints in order to demonstrate the adaptability of the proposed method better.
>
> ---
>
> **Q8: Implementation Details**
>
> **A8:** We appreciate your comment. Theorem 1 suggests setting the regularization strengths $\lambda_t$ and $\lambda_{\text{past}}$ on a certain scale, and the dynamic parameter selection in Algorithm 2 is used to tune the constant. One way to speed up this tuning process is to use a smaller pseudo-validation set per each task, as we did in the experiments with $B = 5$, because only the data points in the pseudo-validation set are used in the dynamic parameter selection procedure and incur additional computational cost. We will also investigate how varying the size of the validation set affects the efficiency of the method.

---

### Official Review · Reviewer_dsCv · 2025-10-28

**Soundness:** 2
**Presentation:** 2
**Contribution:** 2
**Rating:** 4
**Confidence:** 4

**Summary:**

This paper tackles the problem of online continual learning under real concept drift — i.e., when the relationship between inputs and labels evolves over time — rather than the more common “task-incremental” setting in which tasks are static and well-defined. Their main contributions are:


* The authors highlight that many continual learning (CL) works assume a fixed task boundary or static relationship, whereas in many real-world scenarios the underlying model “drifts” as new data arrive (inputs → outputs mapping changes).

* To address concept drift, the authors propose a regularization-based methodology that uses a memory buffer of past examples and constrains the current model’s estimate jointly with past task estimates, under the assumption of a “common center” for the evolving true models. (i.e., current and past models cluster around a latent center)


* In order to adapt to the variability between tasks (or time-segments), the algorithm dynamically tunes task‐specific regularization parameters in the online setting — enabling a better balance of plasticity vs. stability under drift.

* The authors derive an error bound for their estimator, explicitly characterizing how performance depends on: task-relatedness (distance to the latent center), memory buffer size, and regularization strength.

**Strengths:**

* The focus on “real concept drift” rather than idealised fixed-task CL is  relevant for practical deployments (e.g., streaming data, non-stationary domains).


* The derivation of an error bound is a strong component; it connects method design (memory size, regularization) to performance guarantees.


* The dynamic tuning of regularization strength per time‐segment is a meaningful advance over fixed hyper-parameters, enabling the method to adapt to varying drift severity.

**Weaknesses:**

* A key assumption is that the evolving true models (over time) share a “common center” around which they drift. While convenient analytically, this may be unrealistic in many settings where the drift is large or the underlying tasks change drastically. The paper could benefit from discussion or analysis of what happens when the assumption fails.


* If the drift is very abrupt (i.e., the new model is far from the previous center) or tasks are entirely unrelated, it is unclear how well the method will perform. The paper may not sufficiently explore worst‐case drift scenarios.



* The experiment's evaluations are limited to a few synthetic or small‐scale drift settings; it may raise questions about how the method generalises to more complex domains (e.g., large-scale dataset, vision transformer model). The paper may not explore this breadth fully.

**Questions:**

* Could you comment on scenarios where the “true models” do not cluster around a common center (for example, when the drift jumps to a new regime far from previous ones)? How does your algorithm behave in such cases?



* In your experiments, what model sizes/dimensions and what streaming rates (data per time unit) did you consider? How would your method scale to large models, long sequence of time‐segments, or high data throughput?

---

> ### Author Response · Authors · 2025-11-29
>
> We thank the reviewer for the careful review and constructive feedback. Below, we address each point in detail.
>
> ---
>
> **Q1: The authors assume that evolving true models (over time) share a common center around which they drift. These assumptions may be unrealistic in many settings where the drift is large or the underlying tasks change drastically. The paper should discuss whether the proposed method will work if the assumptions fail.**
>
> **A1:**  We appreciate your suggestion. As discussed in the paper, any sequence of $T$ tasks automatically satisfies $(0, \max_{t\in[T]} || \omega_t^* ||_2)$-relatedness, even if there is no clear structure. Larger values of $\varepsilon$ and $\delta$ simply indicate stronger drift. When the drift becomes extremely large, learning each task independently can even be preferable, since forcing joint learning in such cases may lead to negative transfer.
>
> The real-world experiment in Example 5.3 shows a setting where the dependencies between tasks are more complicated than in our synthetic study, which assumes a single shared center. Extending our method to handle multiple centers or a low-rank structure would be an interesting direction for future work, though it is outside the scope of this paper.
>
> ---
>
> **Q2: If the drift is very abrupt (i.e., the new model is far from the previous center) or tasks are entirely unrelated, it is unclear how well the method will perform. The paper may not sufficiently explore worst‐case drift scenarios. Providing deeper analysis would strengthen the paper contributions.**
>
> **A2:** Thank you for the thoughtful comment. We agree that very abrupt drift and completely unrelated tasks are challenging scenarios. Our method is intended to handle moderate but meaningful shifts by anchoring updates around a shared center. When the drift becomes extreme, the benefit of this shared structure naturally weakens, and learning tasks independently may be more appropriate. That said, we agree that evaluating more severe or abrupt drift scenarios would make the paper more complete, and we will address this in the revision.
>
> ---
>
> **Q3: The experiment's evaluations are limited to a few synthetic or small‐scale drift settings; it would be better to evaluate how the method generalises to more complex domains (e.g., large-scale dataset, vision transformer model). This would strengthen the contributions.**
>
> **A3:** Thank you for the suggestion. Evaluating the method on larger and more complex domains, as well as under more extreme drift scenarios, would further strengthen the contributions. To address this, we plan to increase the number of tasks tested and include a discussion on the method’s potential generalization to larger-scale settings in the revision.
>
> ---
>
> **Q4: Could you comment on scenarios where the “true models” do not cluster around a common center? How does your algorithm behave in such cases?**
>
> **A4:** When the “true models” do not cluster around a common center, the benefit of shared-center regularization is reduced. The dynamic tuning parameters $\lambda_t$ and $\lambda_{\text{past}}$ can be automatically chosen smaller, allowing the method to more flexibly accommodate task-specific deviations, though this naturally increases forgetting error. As mentioned earlier, extending the method to handle multiple centers or a low-rank structure as the potential shared pattern is an interesting direction for future work.
>
> ---
>
> **Q5: In your experiments, how would your method scale to large models, large datasets and long task sequence?**
>
> **A5:** Thanks for the question. Our method is pretty efficient since it only updates the shared center and task-specific parameters at each step, which makes it easier to scale to longer task sequences. For larger models or datasets, most of the work comes from the usual model updates. We’ve focused on medium-scale experiments in this paper, but the method should also work for larger settings.

---

### Official Review · Reviewer_P1H5 · 2025-10-30

**Soundness:** 2
**Presentation:** 3
**Contribution:** 3
**Rating:** 6
**Confidence:** 4

**Summary:**

The paper studies online continual learning when the input stream stays roughly the same but the label rule changes over time (“concept drift”). The authors assume that most tasks are actually very similar to one hidden “center” model, and only a small portion can be different. On top of that, they propose an objective that learns from the current data, from a replay buffer, and at the same time pulls everything toward the shared center. They also add an online step that, at the start of each task, tries a few regularization strengths and picks the one that fits this task best. They give a generalization bound under this “all tasks are close to one center” setting, and they show experiments on synthetic data and on a medical dataset where the method beats common continual-learning baselines.

**Strengths:**

1.Clear and tidy formulation that connects replay-style and regularization-style continual learning.

2.Explicit assumption about how similar tasks are, which many CL papers do not write down.

3.Practical online tuning step, so we do not have to hand-tune for each task.

4.Analysis that clearly shows how task similarity, buffer size, and regularization strength matter.

**Weaknesses:**

1.The intro sounds general, but the method and theory only work when almost all tasks are variants of the same model.

2.The key assumption appears too late in the paper.

3.The objective keeps three sets of parameters, even though in the end we only keep the shared one; this needs a clearer justification or an ablation.

4.The weights between old data and new data only depend on how many samples we have, not on how different they are, which is odd for drift.

5.The online tuning step seems to assume we know when a new task starts.

6.The replay buffer is “blind” to the shared center; a center-aware buffer would match the story better.

7.Theory is for the easiest case (all tasks close), not for the mixed case (some tasks far).

8.Synthetic experiments are built exactly the way the assumption says, so the results are a bit circular.

9.No baselines from the concept-drift community.

**Questions:**

1.You claim “real concept drift,” yet assume a single latent center with (ε,δ)-related tasks. Are you actually studying center-constrained drift? If so, shouldn’t the title/intro say so explicitly, and why is this subclass representative of “real” drift in practice?

2.Why is the central (ε,δ)-related assumption only introduced in Sec. 3? Could you front-load it in Sec. 1 and state applicability limits—e.g., which common drifts (directional, periodic, multi-center) are out of scope?

3.Relative to OMTL/Lifelong learning with shared structure, is the novelty the integration (shared center + replay + online tuning) or a new form of sharing itself? Can you position this explicitly to avoid over-claiming originality?

4.If the final output is θ, why not optimize a θ-only objective that blends past and current losses? What is the operational gain of explicit ω_past and ω_t—different λ/optimizers only, or real performance gains? Any θ-only ablation?

5.Under strong drift, shouldn’t small fresh data outweigh large stale data? Why are a1,a2 purely count-based instead of similarity/drift-aware? Can Sec. 3.2’s pseudo-validation scores feed into (a1,a2) for a coherent stability-plasticity trade-off?

6.Alg. 2 uses the first B samples as pseudo-validation—does this assume known task boundaries? If boundaries are unknown or intra-task drift exists, do you need sliding windows/change-point tests/periodic re-tuning? What’s the runtime vs. |Λ| and B?

7.If the core idea is contraction toward θ, why keep θ-agnostic reservoir sampling? Could you test θ-aware buffering (e.g., gradient alignment, center/outlier discriminability, core-set selection) and show effects on the stability–plasticity trade-off?

8.The main bound targets ε=0. For ε>0, how do errors scale with ε, δ, M, λ? Can you provide a relaxed bound or at least a “graceful degradation” analysis (to which baseline do we regress, up to what ε is it robust)?

9.Synthetic setups mirror the single-center assumption. Under directional drift, two-center switching, periodic re-occurrence, or subspace-local drift, do you still beat ER/EWC/AGEM? Any stress tests that violate the assumption and profile failure modes?

---

> ### Author Response · Authors · 2025-11-29
> **Responses (Part 1/3)**
>
> We thank the reviewer for the careful review and constructive feedback. Below, we address each point in detail.
>
> ---
>
> **Q1: The intro sounds general, but the method and theory only work when almost all tasks are variants of the same model. The key assumption appears too late in the paper.**
>
> **A1:** Thank you for the comment. Assumption 1 is quite general;  any sequence of $T$ tasks automatically satisfies $(0, \max_{t\in[T]} || \omega_t^* ||_2)$-relatedness, even when no clear structure is present. Our method is specifically designed to leverage a shared single center across tasks, and extending it to settings with multiple centers or a low-rank structure would be an interesting direction for future work. We will make the scope and intended use of the method more explicit in the Introduction.
>
> ---
>
> **Q2: The objective keeps three sets of parameters, even though in the end we only keep the shared one; this needs a clearer justification or an ablation.**
>
> **A2:** In continual learning, the goal is not to produce an accurate model for a single task. From Algorithm 1, we can see that the estimates of the current and past are intermediate quantities that are not output, and are only used to update the estimate $\hat{\theta}$ that we care about. If the objective in (2) were optimized only with respect to $\theta$, it would reduce to the traditional experience replay (ER) method [1], whose empirical performance is weaker than our method, as shown in our experiments.
>
> ---
>
> **Q3: The weights between old data and new data only depend on how many samples we have, not on how different they are, which is odd for drift.**
>
> **A3:** The reason is that we aim to treat every task fairly, weighting them based on their sample sizes to minimize overall generalization error. We acknowledge that this approach works best when the drift between tasks is moderate. In cases of severe drift, we recommend using drift detection techniques to identify and remove tasks that are far from the others.
>
> ---
>
> **Q4: The online tuning step seems to assume we know when a new task starts.**
>
> **A4:** Yes, our method is not task-free, meaning that task boundary information is required and we know when a new task begins. This limitation is discussed in Section A.3 of the Appendix.
>
> ---
>
> **Q5: The replay buffer is “blind” to the shared center; a center-aware buffer would match the story better.**
>
> **A5:** Thank you for the insightful suggestion. In our current design, the replay buffer is intentionally kept simple using reservoir sampling and does not take the shared center into account. Considering how stored samples relate to the center is an interesting direction for future improvement, and we will highlight this as a possible extension.
>
> ---
>
> **Q6: Theory is for the easiest case (all tasks close), not for the mixed case (some tasks far).**
>
> **A6:**  The theory is established under the assumption $\varepsilon = 0$, but this does not imply that all tasks are identical. The parameter $\delta$ still controls the distance between the true models of different tasks.
>
> ---
>
> **Q7: Synthetic experiments are built exactly the way the assumption says, so the results are a bit circular.**
>
> **A7:**  The synthetic settings are designed to match the task-similarity structure assumed in the theory, mainly to show that the proposed method performs well when the “ideal’’ assumptions hold. We also include experiments (Example 5.2) where the assumptions in Section 4 are not strictly satisfied, and the method still performs competitively. In the real-world example (Example 5.3), we cannot verify whether the assumptions hold, yet the method achieves smaller generalization and forgetting error. Taken together, these results indicate that the method remains effective beyond the idealized theoretical setting.
>
> ---
>
> **Q8: No baselines from the concept-drift community.**
>
> **A8:** Thank you for the comment. We agree that comparison with concept drift methodologies is important. In the continual learning literature, most existing methods focus on preventing forgetting under covariate drift or class incremental scenario, rather than explicitly addressing concept drift, where the underlying relationship between inputs and labels changes while the input distribution may remain the same.
>
> ---
>
> **Q9: You claim “real concept drift,” yet assume a single latent center with $(\varepsilon, \delta)$-related tasks. Are you actually studying center-constrained drift? If so, shouldn’t the title/intro say so explicitly, and why is this subclass representative of “real” drift in practice?**
>
> **A9:** Please refer to our response to Q1 above.
>
> ---
> [1]Chaudhry, A., Rohrbach, M., Elhoseiny, M., Ajanthan, T., Dokania, P., Torr, P. and Ranzato, M., 2019, June. Continual learning with tiny episodic memories. In Workshop on Multi-Task and Lifelong Reinforcement Learning.

---

> ### Author Response · Authors · 2025-11-29
> **Responses (Part 2/3)**
>
> **Q10: Why is the central $(\varepsilon, \delta)$-related assumption only introduced in Sec. 3? Could you front-load it in Sec. 1 and state applicability limits—e.g., which common drifts (directional, periodic, multi-center) are out of scope?**
>
> **A10:** Please refer to our response to Q1 above.
>
>
> ---
>
> **Q11: Relative to OMTL/Lifelong learning with shared structure, is the novelty the integration (shared center + replay + online tuning) or a new form of sharing itself? Can you position this explicitly to avoid over-claiming originality?**
>
> **A11:** Thank you for the question. As noted in the Introduction, the continual learning literature has not carefully addressed concept drift, especially in terms of giving a clear theoretical description of how tasks relate under drift. Our contribution is not a new form of sharing by itself. The novelty comes from how we bring several pieces together and analyze them under a setting that captures concept drift.
>
> The method can be viewed as a modified version of regularization and memory-based ideas in continual learning, but we integrate a shared center, replay, and online tuning into one framework that is supported by theory. The shared center is used because many real drift patterns can be approximated by movement around a common reference point. The online parameter tuning is developed as an intermediate step so the regularization strength can adapt as tasks arrive.
>
> Although the method is built to leverage a shared center, the experiments show that when there is a meaningful amount of concept drift, it still offers clear advantages over approaches that do not account for such drift. We will make this positioning more explicit to avoid overstating the originality of the contribution.
>
> ---
>
> **Q12: If the final output is $\theta$, why not optimize a $\theta$-only objective that blends past and current losses? What is the operational gain of explicit $\omega_{\text{past}}$ and $\omega_t$—different $\lambda$/optimizers only, or real performance gains? Any $\theta$-only ablation?**
>
> **A12:** Please refer to our response to Q2 above.
>
> ---
>
>
> **Q13: Under strong drift, shouldn’t small fresh data outweigh large stale data? Why are $a_1, a_2$ purely count-based instead of similarity/drift-aware? Can Sec. 3.2’s pseudo-validation scores feed into $(a_1, a_2)$  for a coherent stability-plasticity trade-off?**
>
> **A13:** Please refer to our response to Q3 above. It is indeed possible to tune the $(a_1, a_2)$ parameters in Algorithm 2, but doing so would substantially increase the computational burden, as the size of the candidate set would be multiplied by the number of possible $(a_1, a_2)$ options.
>
> ---
>
>
> **Q14: Alg. 2 uses the first B samples as pseudo-validation—does this assume known task boundaries? If boundaries are unknown or intra-task drift exists, do you need sliding windows/change-point tests/periodic re-tuning? What’s the runtime vs. $|\Lambda|$ and $B$?**
>
> **A14:**  Our method is not task-free, so it requires knowledge of task boundaries and assumes we know when a new task begins. We will add experiments to evaluate how the test batch size $B$ and the size of the candidate set $s_\lambda$ affect both running time and estimation error.

---

> ### Author Response · Authors · 2025-11-29
> **Responses (Part 3/3)**
>
> **Q15: If the core idea is contraction toward θ, why keep $\theta$-agnostic reservoir sampling? Could you test $\theta$-aware buffering (e.g., gradient alignment, center/outlier discriminability, core-set selection) and show effects on the stability–plasticity trade-off?**
>
> **A15:** Thank you for the insightful suggestion. We keep the reservoir sampling $\theta$-agnostic to maintain simplicity and avoid bias from an imperfect estimate of the shared center. This ensures the buffer can still store a representative set of past tasks even when $\theta$ is not well-estimated at the beginning. We agree that exploring $\theta$-aware buffering strategies, such as gradient alignment as proposed in Aljundi et al. [2], could be interesting and might improve the stability and plasticity balance, and we plan to test this in future experiments.
>
> ---
>
> **Q16: The main bound targets $\varepsilon = 0$. For $\varepsilon > 0$, how do errors scale with $\varepsilon, \delta, M, \lambda$? Can you provide a relaxed bound or at least a “graceful degradation” analysis (to which baseline do we regress, up to what $\varepsilon$ is it robust)?**
>
> **A16:** Thank you for the constructive suggestion. We will consider the more general case when $\varepsilon > 0$ and provide additional discussion on its implications.
>
> ---
>
> **Q17: Synthetic setups mirror the single-center assumption. Under directional drift, two-center switching, periodic re-occurrence, or subspace-local drift, do you still beat ER/EWC/AGEM? Any stress tests that violate the assumption and profile failure modes?**
>
> **A17:** We appreciate your suggestion. We plan to include such experiments in future work to better understand the robustness of our approach compared with baselines.
>
> ---
>
> [2] Aljundi, R., Lin, M., Goujaud, B. and Bengio, Y., 2019. Gradient based sample selection for online continual learning. Advances in neural information processing systems, 32.

---

### Meta-Review · Area_Chair_to4z · 2026-01-05

**Summary:**

This paper received four reviews, one with an initial rating of 6 (= marginally above the acceptance threshold), two with 4 (= marginally below the acceptance threshold) and one with 2 (= reject, not good enough). The authors provided a rebuttal to each of the four reviews, but unfortunately the reviewers were not able to respond to this rebuttal as it was only posted after the discussion period had prematurely ended.

As discussed in more detail below, my impression is that the author rebuttal fails to convincingly address the concerns raised by the reviewers, and my expectation is that none of the reviewers would have raised their score in response to this rebuttal. I therefore recommend rejection.

**Reviewer Concerns:**

As discussed in more detail below, a number of concerns were raised by the reviewers that have not yet been convincingly addressed. For example, there are concerns about the limited novelty of the work (reviewer YpUb) and the not clearly stating of the assumption that task boundaries are known (reviewer P1H5). There is also an important concern about the simplifying assumption made in the paper that all tasks share a “common center” (reviewers P1H5 and dsCv), and there is a concern about the small-scale and synthetic nature of experiments (reviewers dsCv and s2AM).

**Reviewer Scores:**

Reviewer P1H5, despite giving the highest rating (= 6) and thus marginally supporting acceptance, still raised a relatively long list of weaknesses and questions. In the rebuttal, rather short answers are provided to these, and in several cases the rebuttal does not really address the concern (for example: the fact that the proposed method assumes knowledge of task boundaries is mentioned only in the Appendix, and no baselines from the concept-drift community are considered). I expect that reviewer P1H5 would not have raised their score in response to this rebuttal.

Reviewer YpUb gave the lowest rating (=2), raising several concerns including limited novelty and issues with both the theoretical and empirical results. I find in particular the authors’ rebuttal in response to the limited novelty unconvincing, with the authors simply indicating that they will expand the related-work section accordingly, without specifying exactly how they will do so. I expect that reviewer YpUb would not have raised their score in response to this rebuttal.

Reviewers dsCv and s2AM both gave an initial rating of 4. Similar to reviewer P1H5, reviewer dsCv also raises the issue that the method proposed in this paper assumes a common center for all tasks, indicating that this is a somewhat limiting assumption. In response, the authors agree that this is a limiting assumption. They say that it is possible to extend their method to more general settings, but they do not provide details in that regard. Both reviewers dsCv and s2AM further raise the issue that the experiments are limited to small-scale and synthetic settings. The authors indicate that in the future they plan to include larger scale experiments. I expect that reviewers dsCv and s2AM would not have raised their score in response to this rebuttal.

---

### Decision · Program_Chairs · 2026-01-26

Reject